# Recovering Policy-Induced Errors:
# Benchmarking and Trajectory Synthesis for Robust GUI Agents

**Tianpeng Bu** [* 1]  **Xin Liu** [* 1]  **Qihua Chen** [* 1]  **Hao Jiang** [1]  **Shurui Li** [1]  **Hongtao Duan** [1]  **Lu Jiang** [1]  **Lulu Hu** [1]
**Bin Yang** [1]  **Minying Zhang** [1]

## Abstract

While GUI agents have advanced rapidly, they often lack the robustness to recover from their own errors, hindering real-world deployment. To bridge this gap at both the evaluation and data levels, we introduce GUI-RobustEval and propose **_Ro_**bustness-driven **_T_**rajectory **_S_**ynthesis. GUI-RobustEval contains 1,216 executable test cases that systematically measure error recovery capabilities across a broad and realistic spectrum of error modes. At the data level, RoTS is a scalable synthesis framework that creates $800k$ high-quality data via a tree-based pipeline that proactively discovers diverse error modes and synthesizes corresponding recovery steps. Our two models, RoTS-7B and RoTS-32B, fine-tuned on our dataset, both demonstrate significant gains on GUI-RobustEval and traditional GUI benchmarks. Notably, RoTS-32B achieves state-of-the-art performance on OSWorld, with a $47.4\%$ success rate and a $33.8\%$ All-Pass@4 score, suggesting that improved long-horizon error recovery ability contributes to both robustness and overall performance. Our code is available at https://github.com/AlibabaResearch/RoTS.

## 1. Introduction

Graphical User Interface (GUI) agents (Hu et al., 2025) have shown impressive progress in automating digital devices, catalyzed by the recent advancements in Vision-Language Models (VLMs) (Team et al., 2024; Hurst et al., 2024; Wang et al., 2024b). However, in real-world deployment, agents frequently make *policy-induced errors* (mistakes generated by the agent's own actions during execution, *e.g.*, incorrect grounding, misinterpretation of the screen state or a wrong

subgoal), trapping the agent in an erroneous state and ultimately causing failure. Therefore, *robustness*, the ability to detect such erroneous states and ultimately complete the task, is crucial for practical deployment (Fig. 1).

However, robustness to policy-induced errors remains underemphasized in both evaluation and training. Existing benchmarks mainly focus on grounding accuracy (Cheng et al., 2024; Li et al., 2025), planning ability (Zheng et al., 2025) and overall task success (Xie et al., 2024; Bonatti et al., 2024). Benchmarks for robustness focus on injected noise (Yang et al., 2025b; Zhao et al., 2025) and adversarial attacks (Liao et al., 2025), but provide limited fine-grained metrics that directly measure error detection and long-horizon recovery from policy-induced mistakes. On the learning side, agents are trained by supervised fine-tuning on GUI trajectories (Xu et al., 2025b; Wu et al., 2025c) or online reinforcement learning (Yang et al., 2025a; Lu et al., 2025a). When reflection-related data is used, it is often manually generated or augmented with offline data (Wang et al., 2025; Wu et al., 2025a; Wanyan et al., 2025), which introduces bias in error types and horizons. Meanwhile, agent frameworks (Wu et al., 2025b; Wang et al., 2024a; Agashe et al., 2025) typically improve robustness by adding reflection and backtracking sub-agents, rather than addressing policy-induced error detection and recovery at the training level.

This leads to two gaps in evaluation and training. (1) **Error-coverage mismatch:** benchmarks and training data over-represent low-level, human-curated errors, while policy-induced failures are often compositional and high-level, leaving agents miscalibrated to realistic failure modes (Fig. 3(a)). (2) **Error-horizon mismatch:** existing evaluation protocols and reflection data are mostly short-horizon (*e.g.*, invalid clicks), while many policy-induced errors only emerge after multiple steps and require long-horizon backtracking and recovery (Fig. 3(b)). This gap motivates two contributions of our work (Fig. 2).

First, we introduce GUI-RobustEval, a benchmark to measure how GUI agents detect and recover from policy-induced errors. GUI-RobustEval has 1216 test cases covering 11 representative error types across 4 controllable error

---

*Equal contribution [1]Alibaba Cloud Computing. Correspondence to: Minying Zhang <minying.zhang@alibaba-inc.com>.

*Proceedings of the $43^{rd}$ International Conference on Machine Learning*, Seoul, South Korea. PMLR 306, 2026. Copyright 2026 by the author(s).

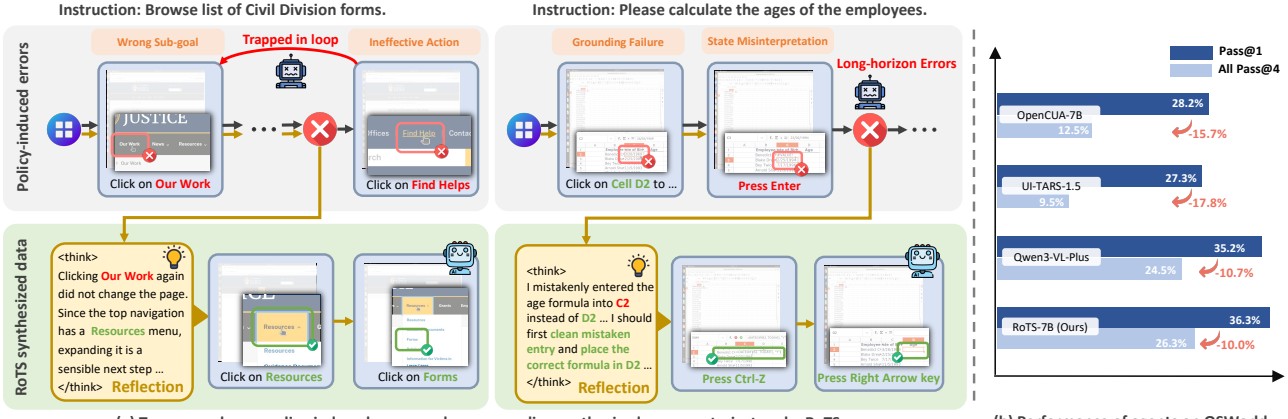

*Figure 1.* Policy-induced errors exhibit diverse types and delayed error detectability. GUI agents struggle to identify and recover from such errors (upper part of Fig. (a)), while RoTS improves this by synthesizing reflection-related data matching policy-induced error distribution (lower part of Fig. (a)). Benefit from this, RoTS achieves lower accuracy drop on All-Pass@4 (Fig. (b)) compared with other methods.

depths. For each task, we provide an erroneous prefix trajectory, reset the environment and the agent to a targeted state at error depth $d$ (*i.e.*, $d$ steps after the root-cause action in the prefix), and ask the agent to take over and finish the task. We report Error-Awareness Rate and Post-Error Success Rate, and analyze both metrics with respect to error types and error depth. These fine-grained metrics provide targeted diagnostics beyond overall success, revealing the type of errors agents fail to recognize and how recovery degrades with increasing depth.

Second, we propose **_Ro_**bustness-driven **_T_**rajectory **_S_**ynthesis (RoTS), a tree-based online data synthesis framework, designed to close the coverage and horizon gaps in policy-induced errors. It iteratively grows a trajectory tree in the GUI environment: on the successful branches, it branches out from fragile states to proactively discover new failure modes while reusing the correct prefix; on the failed branches, it replays from the error state and synthesizes recovery rollouts to produce long-horizon failure-recovery trajectories for training. The two branches respectively close the coverage gap by exposing diverse failure modes and the horizon gap by generating long-horizon recoveries.

In summary, our contributions are: (1) GUI-RobustEval, a benchmark for policy-induced errors that evaluates error awareness and recovery; (2) RoTS, a data synthesis pipeline for generating diverse long-horizon failure-recovery trajectories, together with an 800k-sample dataset and a fine-tuned QWEN2.5-VL that improve robustness on GUI-RobustEval and task success on OSWorld and WindowsAgentArena.

**Conflict of Interest Disclosure.** All authors are employees of Alibaba Cloud Computing. The QWEN-VL series models and RoTS evaluated in this paper were developed at Alibaba Cloud Computing.

## 2. Benchmark for Policy-Induced Errors

### 2.1. Problem Formulation

GUI agent tasks involve a policy $\pi_\theta$ sequentially interacting with a GUI. Following prior work (Nguyen et al., 2025), we model the environment as a POMDP $(\mathcal{U}, \mathcal{A}, \mathcal{S}, \mathcal{O}, \mathcal{T}, \mathcal{R})$. In our setting, $\mathcal{U}$ and $\mathcal{A}$ are natural-language task instructions and actions, $\mathcal{O}$ are screenshots, $\mathcal{S}$ (state) and $\mathcal{T}$ (transition) are determined by the GUI environment, $\pi_\theta$ is instantiated by a VLM, and $\mathcal{R}$ is rule- or VLM-based reward model. At step $i$, the agent samples an action $a_i \sim \pi_\theta(\cdot \mid u, o_i, h_{i-1})$, where $h_{i-1} = (o_1, a_1, \ldots, o_{i-1}, a_{i-1})$. The environment transitions $s_{i+1} \sim \mathcal{T}(s_i, a_i)$ and returns $o_{i+1}$. The process ends when the task is completed or a step limit is reached, producing the full trajectory $\tau$.

### 2.2. GUI-RobustEval

**Revisiting the Policy-Induced Errors.** To understand policy-induced errors, we analyze the error types and error horizon using trajectories from 12 state-of-the-art agents on OSWorld (Xie et al., 2024). In this paper, we define error type as the category of the root-cause action, and error horizon as the minimal number of steps after the root cause required for the error to become identifiable.

We collect $1.5k$ trajectories and use a VLM to annotate error types and visualize their distribution with t-SNE (Fig. 3(a)). For error horizon, experienced annotators label on 300 failed trajectories, the earliest step where the root-cause becomes identifiable (Fig. 3(b)). We apply the same procedure on three representative training datasets: AgentTrek (Xu et al., 2025a) and AgentNet (Wang et al., 2025) (human demonstrations), and GUI-Reflection (Wu et al., 2025a) (offline-augmented reflection data). By comparing these distributions between inference failures and training data, we iden-

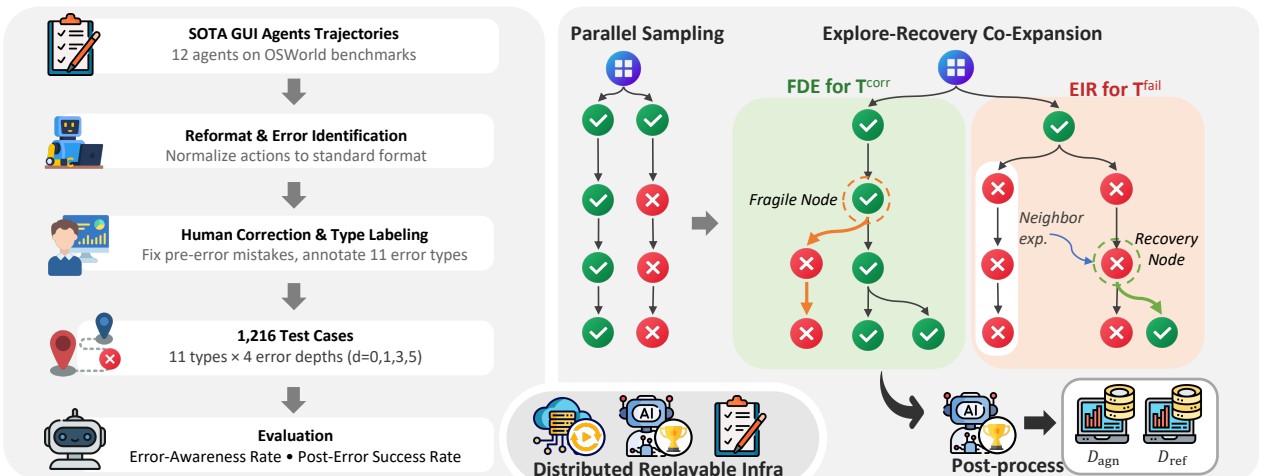

*Figure 2.* Overview of our method. It includes (i) the pipeline for constructing our benchmark, GUI-RobustEval, and (ii) RoTS, the pipeline for synthesizing diverse error-recovery trajectories that cover the policy-induced error distribution. We also build a highly parallel infrastructure that supports high-throughput evaluation and data synthesis.

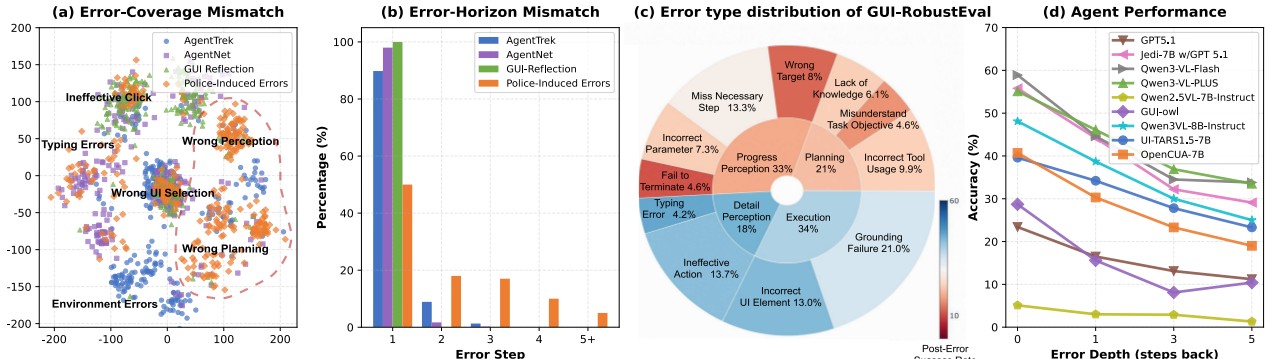

*Figure 3.* (a): Error type distribution of policy-induced errors and existing datasets. (b): Error-horizon distribution of policy-induced errors and existing datasets. (c): Error type percentage in GUI-RobustEval, which is colored by post-error success rate. (d): The post-error success rate w.r.t. the error depth of SOTA agents on GUI-RobustEval.

tify two gaps, *i.e.*, coverage mismatch: training data concentrates on low-level execution errors or errors frequently made by human, while real failures often involve compositional perception and planning; horizon mismatch: training data is dominated by immediately identifiable errors, while real errors may surface only after several steps. These observations motivate GUI-RobustEval, which covers error types in real-execution and controllable error depth for fine-grained evaluation.

**Benchmark Construction.** From $1.5k$ failed trajectories, human experts locate the root-cause step of each failure and assign an error type (multiple choice), yielding $11$ error types and $4$ error depths ($d \in \{0, 1, 3, 5\}$). To keep the evaluation controlled, experts fix any unrelated mistakes before the root-cause to guarantee an error-free prefix. As different agents use different CoT formats and action spaces, we normalize each step into a standard, executable form

(action summary + PyAutoGUI), and convert it back to each agent's native format at test time. For evaluation at depth $d$, we start from a system snapshot and replay all corrected pre-error steps followed by the root-cause step and the next $d$ steps, then let the agent take over with the injected history. We report (1) Error-Awareness Rate: whether the agent recognizes the error immediately after takeover (judged from its output by VLM); and (2) Post-Error Success Rate: whether it can recover and complete the task.

**Benchmark Summary.** GUI-RobustEval contains 1,216 test cases across 11 error types. It distinguishes itself from prior efforts by focusing on realistic, policy-induced errors rather than synthetic errors or external perturbations, as illustrated in Table 1.

GUI-RobustEval provides the following insights: Fig. 3(c) shows the type distribution of policy-induced errors, with color indicating recovery difficulty (defined as $1-$ post-error

success rate, averaged over five SOTA agents). Planning and progress-perception errors are much harder to recover than low-level execution errors, and are also less covered in existing training data. Fig. 3(d) shows that performance drops as error depth increases, likely because the environment drifts further from the goal and the injected post-error history misleads subsequent decisions. These insights motivate the design of RoTS, which explores diverse failure modes and generates long-horizon recovery trajectories.

*Table 1.* Comparison with existing GUI Agent Benchmarks.

| Benchmark | Test Cases | Online | Robustness |
|---|---|---|---|
| OSWorld | 369 | ✓ | - |
| GUI-Reflection | 1,626 | ✗ | Synthetic Error |
| GUI-Robust | 5,318 | ✗ | Environment Disturb |
| D-GARA | 152 | ✓ | Environment Disturb |
| RedTeamCUA | 864 | ✓ | Adversarial Attack |
| GUI-RobustEval | 1216 | ✓ | Policy-Induced |

## 3. *Ro*bustness-driven *T*rajectory *S*ynthesis

### 3.1. Environment Preparation

In this work, following OSWorld (Xie et al., 2024) and WindowsAgentArena (Bonatti et al., 2024), we host Ubuntu and Windows systems on the cloud to achieve high-throughput sampling. We curate $20k$ tasks, all with a reproducible system snapshot, task-related materials and initialization configuration. We adopt WebJudge (Xue et al., 2025) as the outcome reward model $\mathcal{R}$ to assess task-completion correctness. Additionally, we employ a progress critic $\mathcal{R}_p$ model and an action critic $\mathcal{R}_a$ model to evaluate the planning and step-level execution correctness during rollout. To validate these LLM-as-judge components, we conduct a human-agreement study in which these critic models reach 90%, 88.7%, and 90.6% agreement with human annotators, respectively, confirming their reliability for our pipeline (Appendix B.3).

We design the environment so that any node in the trajectory tree is replayable in principle: its underlying state can be restored by replaying the unique root-to-node action prefix on the initial snapshot. To improve replay fidelity, we disable reproducibility-breaking factors (*e.g.*, automatic updates, notification daemons) in the environment. For GUI-RobustEval evaluation, we further enforce strict replayability: each test case is constructed from a *verified* prefix whose replay consistency has been confirmed by human annotators, ensuring that the injected erroneous state is faithfully reproduced at evaluation time. For more details about the infrastructure, task construction and reward model, please refer to Appendix B.

---

**Algorithm 1** Explore–Recovery Co-Expansion

**Require:** Task $u \in \mathcal{U}$; policy model $\pi_\theta$; parallel sampling $N$; rounds $K$; reward model $\mathcal{R}$; progress critic $\mathcal{R}_p$; reflector $\pi_\theta^{er}$; recovery actor $\pi_\theta^{rec}$.

**Ensure:** Expanded tree $T$.

1: $o_1 \leftarrow$ INITENV$(u)$;    $T \leftarrow (\{o_1\}, \emptyset, \emptyset)$
2: **for** $n = 1$ to $N$ **do**           ▷ $N$ parallel sampling
3:     $T \leftarrow$ PARALLELROLLOUT$(\pi_\theta, u, o_1, T, \mathcal{R})$
4: **end for**
5: **for** $k = 1$ to $K$ **do**           ▷ Co-expansion
6:     $T^{\text{corr}}, T^{\text{fail}} \leftarrow$ PRUNEBYREWARD$(T, \mathcal{R})$
7:     **if** $\text{Traj}(T^{\text{corr}}) \neq \emptyset$ **then**           ▷ FDE
8:         $T^{\text{corr}} \leftarrow$ CALCSTEPSUCCESS$(T^{\text{corr}}, \mathcal{R}_p)$
9:         $i^* \leftarrow$ UCB$_{\text{f}}(T^{\text{corr}})$           ▷ Eq. (1)
10:         $\tau_{prefix} \leftarrow$ REPLAY$(T, o_1, o_{i^*})$
11:         $T \leftarrow$ ROLLOUT$(\pi_\theta, u, \tau_{prefix}, T, \mathcal{R})$
12:     **end if**
13:     **if** $\text{Traj}(T^{\text{fail}}) \neq \emptyset$ **then**           ▷ EIR
14:         $T^{\text{fail}} \leftarrow$ ERRORLOCALIZATION$(T, T^{\text{fail}}, \pi_\theta^{er})$
15:         $i^* \leftarrow$ UCB$_{\text{r}}(T^{\text{fail}})$           ▷ Eq. (5)
16:         $\tau_{prefix} \leftarrow$ REPLAY$(T, o_1, o_{i^*})$
17:         $T \leftarrow$ ROLLOUT$(\pi_\theta^{rec}, u, \tau_{prefix}, T, \mathcal{R}; g_{i^*})$
18:     **end if**
19: **end for**
20: **return** $T$

---

### 3.2. Explore-Recovery Co-Expansion

We organize all sampled rollouts for each task into a replayable trajectory tree $T = (O, A, E)$, where nodes are screenshot observations and edges are actions. The environment is initialized from a task-specific fixed configuration, yielding a consistent initial observation.

Building on this structure, as shown in Fig. 2, we perform explore–recovery co-expansion: after $N$ parallel rollouts with policy $\pi_\theta$, we iterate for $K$ rounds, partitioning $T$ via the reward model $\mathcal{R}$ into a successful subtree $T^{\text{corr}}$ and a failed subtree $T^{\text{fail}}$. We then expand both sides: (i) fragility-driven exploration (FDE) uses a progress critic $\mathcal{R}_p$ and a UCB-style rule to select high-fragility nodes in $T^{\text{corr}}$, replays to the selected node, and continues rollout with $\pi_\theta$; (ii) experience-informed recovery (EIR) uses a reflector $\pi_\theta^{er}$ to localize error states in $T^{\text{fail}}$ and derive advice from neighboring branches, prioritizes error nodes via UCB, and launches advice-conditioned recovery rollouts with a recovery actor $\pi_\theta^{rec}$. The overall procedure is summarized in Algorithm 1.

### 3.3. Fragility-Driven Exploration

**Average Step-level Success Rate.** For each node $o_i$ in the correct subtree $T^{\text{corr}}$, we sample $N$ actions from the policy model $\{a_{i,1}, \ldots, a_{i,N}\} \sim \pi_\theta(o_i)$, and use a pre-operative progress critic model (Wanyan et al., 2025) $\mathcal{R}_p$

to predict a binary correctness label for each action: $r_{i,n} \sim \mathcal{R}_p(o_i, a_{i,n}, h_{i-1})$. We then compute the mean correctness: $r_i = \frac{1}{N} \sum_{n=1}^{N} r_{i,n}$. $r_i$ serves as an estimate of the probability that $\pi_\theta$ proposes a correct next action from $o_i$.

**Fragility-Score and Node Selection.** Based on $r_i$, we define the fragility-score for node $o_i$ using a standard UCB criterion to encourage breadth:

$$f_i = (1 - r_i) + c\sqrt{\frac{\ln\left(V_{p(i)}^f + 1\right)}{V_i^f + 1}}, \quad (1)$$

where $V_i^f$ and $V_{p(i)}^f$ are the number of times node $o_i$ and its parent node are expanded by FDE, and $c > 0$ is the exploration constant, which encourages exploring less-visited nodes. Thus, we select the node with the highest fragility-score among the nodes in $T^{\text{corr}}$, i.e.,

$$i^* = \arg\max_i f_i. \quad (2)$$

After selecting $i^*$, we reset the environment and replay the prefix actions $(a_1, \ldots, a_{i^*-1})$ to restore the state $o_{i^*}$, from which the policy model $\pi_\theta$ is deployed to expand the tree.

### 3.4. Experience-Informed Recovery

**Neighbor-Experience Guided Error Localization.** For each trajectory $\tau$, our reward model $\mathcal{R}$ not only outputs the reward but also extracts a reusable trajectory experience $E_\tau$ (details in Appendix B.3.2). To localize failures, for each failed trajectory $\tau^{\text{fail}} \in \text{Traj}(T^{\text{fail}})$, we collect its sibling-branch neighbors in the full tree $T$ and aggregate their experiences into a neighbor-experience set

$$\mathcal{E}(\tau^{\text{fail}}) \triangleq \{E_{\tau^{\text{nb}}} \mid \tau^{\text{nb}} \in \mathcal{N}(\tau^{\text{fail}})\}. \quad (3)$$

Conditioning on $(u, \tau^{\text{fail}}, \mathcal{E}(\tau^{\text{fail}}))$, an experience-informed reflector proposes candidate error steps together with recovery guidance and an expansion priority:

$$\{(i, g_i, p_i)\} \sim \pi_\theta^{er}(u, \tau^{\text{fail}}, \mathcal{E}(\tau^{\text{fail}})). \quad (4)$$

Here $g_i$ denotes the natural-language recovery guidance and $p_i$ the expansion priority for candidate error step $i$. We aggregate proposals across all failed trajectories in $T^{\text{fail}}$ to obtain a global candidate error-state set $\mathcal{I}$ with associated recovery guidance and expansion priority $(g_i, p_i)$.

**Advice-Conditioned Recovery.** Similar to Eq. 1, we select a recovery node using a UCB-style criterion:

$$s_i = p_i + c\sqrt{\frac{\ln\left(V_{p(i)}^r + 1\right)}{V_i^r + 1}}, \qquad i^* = \arg\max_{i \in \mathcal{I}} s_i, \quad (5)$$

where $V_i^r$ and $V_{p(i)}^r$ are the numbers of times node $o_i$ and its parent $o_{p(i)}$ have been selected as the starting nodes for

recovery expansion, and $c > 0$ is the exploration constant. We then restore $o_{i^*}$ and deploy the recovery actor to perform an advice-conditioned rollout:

$$\tau^{\text{rec}} \sim \pi_\theta^{rec}(u, o_{i^*}, h_{i^*-1}, g_{i^*}). \quad (6)$$

### 3.5. Dataset Construction and Training

We collect data via tree-based exploration, which yields branching trajectories with shared prefixes. Moreover, agent rollouts are step-wise noisy: successful trajectories may include incorrect actions, while failed ones may contain correct steps and useful reflections. Environment stochasticity can also cause state-transition inconsistencies during replay, introducing additional noise. Supervising with whole trajectories is therefore inefficient and may introduce noise. Instead, we apply a post-processing pipeline to select correct steps for training. Each training instance is represented as:

$$x_i = \big(u, \; h_{i-1}, \; o_i, \; a_i\big), \quad (7)$$

where $u$ is the instruction, $h_{i-1}$ is the history up to step $i-1$, $o_i$ is the current observation, and $a_i$ is a React-style CoT (Yao et al., 2023) followed by the executable action. During training, we supervise only the tokens in $a_i$, treating history as context to avoid propagating noise from imperfect rollouts.

To achieve this, we first apply VLM-based posterior filtering to discard trajectories with inconsistent state transitions caused by environment stochasticity. We then use the progress critic $\mathcal{R}_p$ (plan consistency) and an action critic $\mathcal{R}_a$ (execution correctness) to remove incorrect steps from both successful and unsuccessful trajectories. This adds little computation overhead by reusing critic outputs from tree expansion and running extra checks only when needed. We further use a VLM-based reflection validator $\mathcal{R}_f$ to split the filtered data into $\mathcal{D}_{\text{agn}}$ and $\mathcal{D}_{\text{ref}}$, representing *reflection-agnostic* and *reflection-related* subsets respectively, depending on whether the step exhibits effective reflection behavior. Finally, we apply rule-based deduplication on these two subsets to obtain the final dataset. Additional details on CoT synthesis, data deduplication, and examples are provided in the Appendix C.3.

Following (Yuan et al., 2025), we train on a mixture of reflection-agnostic and reflection-related subsets:

$$\mathcal{D}_{\text{train}} = \mathcal{D}_{\text{agn}} \cup \lambda_{\text{ref}} \mathcal{D}_{\text{ref}}, \quad (8)$$

where $\lambda_{\text{ref}} \in [0, 1]$ is a hyperparameter controlling the fraction of reflection-related data: we sample from $\mathcal{D}_{\text{agn}}$ and $\mathcal{D}_{\text{ref}}$ so that reflection-related steps comprise a $\lambda_{\text{ref}}$ proportion of $\mathcal{D}_{\text{train}}$. We use teacher forcing with negative log-likelihood:

$$\mathcal{L}(\theta) = \mathbb{E}_{(u,h,o,a) \sim \mathcal{D}_{\text{train}}} \left[-\log \pi_\theta(a \mid u, h, o)\right]. \quad (9)$$

*Table 2.* Evaluation results on GUI-RobustEval of different GUI agents. The best and second best performance for open-sourced models are highlighted by bold and underline.

| Agent Model | Success Rate with Error Depth | | | | Awareness |
|---|---|---|---|---|---|
| | **0** | **1** | **3** | **5** | |
| *Proprietary Models* | | | | | |
| GPT 5.1 | 23.4 | 16.5 | 13.1 | 11.2 ($\downarrow 52\%$) | 33.9 |
| Jedi-7B w/ GPT 5.1 | 55.8 | 44.2 | 32.2 | 29.1 ($\downarrow 48\%$) | 34.6 |
| Qwen3-VL-Flash | 54.9 | 44.6 | 34.5 | 32.1 ($\downarrow 42\%$) | 63.9 |
| Qwen3-VL-Plus | 55.1 | 46.1 | 36.9 | 33.5 ($\downarrow 39\%$) | 65.4 |
| *Open-Source Models* | | | | | |
| Qwen2.5-VL-7B-Instruct | 5.1 | 3.0 | 2.9 | 1.3 ($\downarrow 75\%$) | — |
| GUI-Owl-7B | 28.7 | 15.6 | 8.1 | 10.4 ($\downarrow 64\%$) | 5.9 |
| Qwen3-VL-8B-Instruct | 48.1 | 38.7 | 30.0 | 25.0 ($\downarrow 48\%$) | — |
| UI-TARS1.5-7B | 39.6 | 34.2 | 27.8 | 23.3 ($\downarrow 41\%$) | 38.0 |
| OpenCUA-7B | 40.7 | 30.3 | 23.3 | 19.0 ($\downarrow 53\%$) | 46.3 |
| OpenCUA-32B | 45.5 | 37.2 | 28.6 | 25.9 ($\downarrow 53\%$) | 50.3 |
| **RoTS-7B** | 43.5 | 36.6 | 30.1 | 26.7($\downarrow 38\%$) | 51.9 |
| **RoTS-32B** | **49.7** | **41.8** | **36.5** | **33.2**($\downarrow 33\%$) | **58.8** |

## 4. Experiment

### 4.1. Experimental Setup

**Benchmarks.** We evaluate RoTS on three benchmarks: GUI-RobustEval, OSWorld-Verified (Xie et al., 2024) (369 Ubuntu tasks) and WindowsAgentArena (Bonatti et al., 2024) (154 Windows 11 tasks). On GUI-RobustEval, the error depth $d$ ranges from 0 (verified-correct prefix only) to 5, and each run allows up to 50 steps (including the erroneous prefix); we report error awareness rate and success rate across error depths and types, averaged over 3 independent runs. On traditional benchmarks, our models are tested under 15 and 50 steps, averaged over 4 runs. We also report All-Pass@4 indicating the agent achieves consistent success in all 4 independent runs, measuring robustness. We compare against a suite of strong proprietary and open-sourced GUI agents. Model details, infrastructure, and full benchmark configurations are provided in Appendix D.

**Implementation Details.** We synthesize trajectories on $20k$ online tasks using three GUI agents as policies (OpenCUA-7B, UI-TARS-1.5-7B and Qwen3-VL-Plus). For each task and each policy model, we initialize the tree with $N = 4$ rollouts and run 32 rounds of Explore-Recovery Co-expansion, yielding 68 trajectories per task. Our pipeline supports up to 120 parallel synthesis tasks, with open-weights policies served on 32 A100 GPUs and closed-source models accessed via their official APIs. The $20k$ synthesis tasks are fully disjoint from all evaluation benchmarks (GUI-RobustEval, OSWorld-Verified, and WindowsAgentArena) at both the task and asset level; only base OS snapshots are shared. We experiment with different data mixture strategies by varying $\lambda_{\text{ref}}$ and choose $\lambda_{\text{ref}} = 0.1$. We use total $800k$ training samples containing $720k$ from $\mathcal{D}_{\text{agn}}$ and $80k$ from $\mathcal{D}_{\text{ref}}$ to fine-tune Qwen2.5-VL-7B and Qwen2.5-VL-32B with SFT. Full implementation details are provided in Appendix E.

### 4.2. Main Results

**Results on GUI-RobustEval.** We report two complementary metrics: Error-Awareness Rate measures whether the agent recognizes the error at takeover, while Post-Error Success Rate measures whether it can actually recover and complete the task. As shown in Table 2, the two metrics are positively correlated overall but not equivalent—awareness is a prerequisite for recovery, yet recovery additionally requires re-planning and multi-step execution. For example, GPT 5.1 and Jedi-7B w/ GPT 5.1 show similar awareness ($\sim 34\%$) but diverge sharply in success rate, as Jedi's separated planning-grounding architecture converts similar error perception into more effective recovery actions. Among open-source models, RoTS-7B and RoTS-32B achieve the highest scores on both metrics (awareness 51.9%/58.8%, average success 34.2%/40.3%), surpassing OpenCUA-32B. Moreover, under the most challenging setting (error depth 5), RoTS-7B and RoTS-32B maintain 26.7% and 33.2% success rate with the lowest performance drop, highlighting the advantage of RoTS in both identifying and recovering from policy-induced errors across extended horizons.

**Results on OSWorld.** Table 3 shows even SOTA GUI agents suffer from consistently completing the task under All-Pass@4. OpenCUA-32B achieves 15.5 All-Pass@4, dropping $\downarrow 54.5\%$ compared to averaged success rate. In contrast, RoTS-32B achieves 33.8% on All-Pass@4, which have lower accuracy drop ($\downarrow 28.7\%$), highlighting the advantage of RoTS in enhancing the robustness of GUI agents. Moreover, increasing the max step budget from 15 to 50 yields an additional $+4.6\%$ for RoTS-7B, indicating that longer allowable steps amplify the gains from RoTS's effective reflection. Overall, RoTS-7B and RoTS-32B achieve 36.3% and 47.4% at max step 50, surpassing other open-weights GUI models of comparable scales. From the case studies in Appendix F.4, it can be observed that this strong robustness and performance is largely driven by the enhanced reflection on diverse and long-horizon policy-induced errors. Additionally, the result on WindowsAgentArena is in Table 11.

### 4.3. Ablations

Unless otherwise specified, all ablations and analysis use the same training/evaluation protocol. For training, we fix the total fine-tuning data size to $100k$ and fine-tune Qwen2.5-VL-7B with the same hyperparameters across all variants. We report results on GUI-RobustEval and OSWorld under the same metrics as the main experiment.

**The Effectiveness of Co-Expansion.** We compare different rollout strategies under the same rollout budget on $5k$ online tasks. We evaluate: (1) PS: 36 parallel sampling; (2) PS+FDE: 4 parallel sampling with 64 fragility-

*Table 3.* Comparison of the state-of-the-art methods on the OSWorld benchmark. We report the success rate (%) under maximum step 15 and $\geq 50$ as the evaluation metrics. All-Pass@4 is reported to show the success rate across all 4 independent runs with max steps 50. † indicates our re-implemented results, averaged over 4 runs.

| Agent Method | Data Type | All-Pass@4 (50) | Max Steps: 15 | Max Steps: $\geq$ 50 |
|---|---|---|---|---|
| *Proprietary Models* | | | | |
| OpenAI CUA (OpenAI, 2025a) | In-house | – | 26.0 | 31.3 |
| Doubao-1.5-Thinking (Guo et al., 2025a) | In-house | – | 31.9 | 40.0 |
| Claude 4.5 Sonnet (Anthropic, 2025) | In-house | – | **42.9** | **58.1** |
| Qwen3-VL-Flash (Bai et al., 2025a) | In-house | 22.1 | 32.1† | 41.6 |
| Qwen3-VL-Plus (Bai et al., 2025a) | In-house | 24.5 | 33.1† | 35.2† |
| *Open-Weights Models (Smaller Size)* | | | | |
| UI-TARS-1.5-7B (Qin et al., 2025) | In-house | 9.5 | 24.5 | 27.3 |
| OpenCUA-7B (Wang et al., 2025) | Open-source | 12.5 | 24.3 | 28.2 |
| GUI-OWL-7B (Ye et al., 2025) | In-house | 14.7 | 27.1 | 29.4 |
| Qwen3-VL-8B-Thinking (Bai et al., 2025a) | In-house | 21.6 | 29.2† | 33.9 |
| **RoTS-7B** | Open-source | **26.3** | **31.7**† | **36.3**† |
| *Open-Weights Models (Larger Size)* | | | | |
| Qwen2.5-VL-32B (Bai et al., 2025b) | In-house | 0.7 | 3.0 | 3.9 |
| Qwen2.5-VL-72B (Bai et al., 2025b) | In-house | 1.1 | 4.4 | 5.0 |
| UI-TARS-72B-DPO (Qin et al., 2025) | In-house | 11.0 | 24.0 | 25.8 |
| OpenCUA-32B (Wang et al., 2025) | Open-source | 15.5 | 29.7 | 34.1 |
| Qwen3-VL-32B-Thinking (Bai et al., 2025a) | In-house | 21.1 | 28.1† | 41.0 |
| **RoTS-32B** | Open-source | **33.8** | **42.8**† | **47.4**† |

*Table 4.* Ablation on different rollout strategies under the similar budget. The best scores are highlighted as bold.

| Data Source | GUI-RobustEval(%) | | OSWorld (%) | |
|---|---|---|---|---|
| | Aware. | Post. Succ. | All-Pass@4 | Max Steps: 50 |
| PS | 19.9 | 12.1 | 8.6 | 18.1 |
| + FDE | 22.5 | 14.4 | 9.1 | 19.6 |
| + EIR | 28.3 | 18.1 | 12.1 | 19.5 |
| **+ FDE + EIR** | **32.1** | **22.1** | **14.1** | **21.4** |

**Quality of Our Dataset.** We evaluate our dataset quality by comparing with AgentNet (Wang et al., 2025), a high-quality open-source dataset from human demonstrations. AgentNet contains reflection-related samples reflecting on human-execution errors rather than policy-induced errors. We split AgentNet into reflection-agnostic and reflection-related subsets, denoted as $\mathcal{D}_{\text{agn (hum)}}$ and $\mathcal{D}_{\text{ref (hum)}}$.

We design the following settings: (1) $\mathcal{D}_{\text{agn (hum)}}$: $100k$ reflection-agnostic samples from AgentNet; (2) $\mathcal{D}_{\text{agn (hum)}} \cup \mathcal{D}_{\text{ref (hum)}}$: $90k$ and $10k$ from AgentNet's reflection-agnostic and reflection-related subsets; (3) $\mathcal{D}_{\text{agn (hum)}} \cup \mathcal{D}_{\text{ref}}$: $90k$ from AgentNet and $10k$ from RoTS's reflection-related subset; (4) $\mathcal{D}_{\text{agn}} \cup \mathcal{D}_{\text{ref}}$: $90k$ and $10k$ from RoTS's reflection-agnostic and reflection-related subsets.

Table 5 shows that adding human reflections $\mathcal{D}_{\text{ref (hum)}}$ only brings small gains over $\mathcal{D}_{\text{agn (hum)}}$ (OSWorld All-Pass@4 7.8 to 8.4, success rate 15.3 to 16.1). In contrast, replacing them with our policy-induced reflections yields much larger improvements (OSWorld All-Pass@4 8.4 to 11.6, success rate 16.1 to 18.8). This suggests our reflection data matches the policy-induced error distribution better, leading to more effective reflection and improved robustness. Finally, using fully policy-induced data achieves the best results, *i.e.*, 14.1 and 21.4 for All-Pass@4 and success rate, validating the effectiveness of our dataset.

driven exploration; (3) PS+EIR: 4 parallel sampling with 64 experience-informed recovery; (4) PS+EIR+FDE: 4 sampling with 32 rounds of FDE and EIR respectively. Since our method reuses previously generated prefixes, parallel sampling starts from the root and thus uses fewer parallel branches to match the same budget. For all variants, we sample $90k$ and $10k$ data from $\mathcal{D}_{\text{agn}}$ and $\mathcal{D}_{\text{ref}}$ respectively.

As shown in Table 4, adding FDE brings gains on OSWorld success rate (18.1 to 19.6), while contributing less improvement to robustness (8.6 to 9.1). This suggests using FDE only mainly benefits from the model's self-reflection and error exploration. Adding EIR yields larger robustness gains, raising All-Pass@4 to 12.1. Combining EIR+FDE achieves the best overall performance, reaching 14.1 on All-Pass@4, and 21.4 on OSWorld, validating our co-expansion strategy.

*Table 5.* A study on the data quality under a fixed dataset size of $100k$ samples from different data mixtures.

| Training Data | GUI-RobustEval(%) | | OSWorld (%) | |
|---|---|---|---|---|
| | Aware. | Post. Succ. | All-Pass@4 | Max Steps: 50 |
| $\mathcal{D}_{\text{agn (hum)}}$ | 15.6 | 10.4 | 7.8 | 15.3 |
| $\mathcal{D}_{\text{agn (hum)}} \cup \mathcal{D}_{\text{ref (hum)}}$ | 17.2 | 11.5 | 8.4 | 16.1 |
| $\mathcal{D}_{\text{agn (hum)}} \cup \mathcal{D}_{\text{ref}}$ | 26.6 | 19.5 | 11.6 | 18.8 |
| $\mathcal{D}_{\textbf{agn}} \cup \mathcal{D}_{\textbf{ref}}$ | **32.1** | **22.1** | **14.1** | **21.4** |

## 4.4. Analysis

**Sensitivity to $\lambda_{\text{ref}}$.** We sweep the reflective data ratio $\lambda_{\text{ref}}$ in Equation 8 under the same setup. We fix the dataset size to $100k$, and we set $\lambda_{\text{ref}}=0$, and increase $\lambda_{\text{ref}}$ by progressively replacing a portion of $\mathcal{D}_{\text{agn}}$ with $\mathcal{D}_{\text{ref}}$ while keeping the total number of samples fixed. We evaluate $\lambda_{\text{ref}} \in \{0, 0.05, 0.1, 0.15, 0.2, 0.3\}$ and report the OSWorld results in Fig. 4. It is shown that introducing reflective data improves performance and the robustness over $\lambda_{\text{ref}}=0$, and the best results are achieved at $\lambda_{\text{ref}}=0.1$, reaching $21.4$ and the All-Pass@4 is improved more significantly ($14.1\%$). When further increasing $\lambda_{\text{ref}}$, performance drops, even worse than without reflection data ($14.8\%$ at $\lambda_{\text{ref}} = 0.3$), suggesting that too much reflective data leads to ineffective reflections, hurting the performance. Overall, reflective and reflection-agnostic steps are complementary and should be balanced.

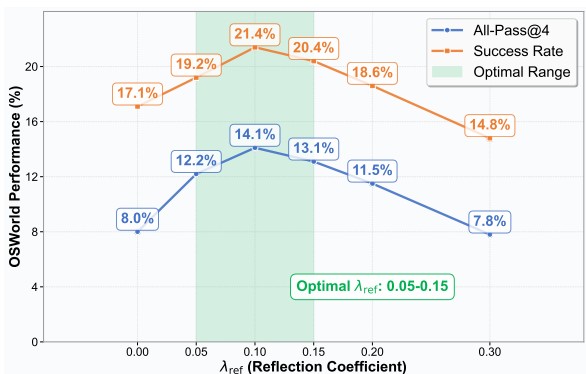

*Figure 4.* The impact of different ratio of reflection data.

**Expansion Rounds and Dataset Size.** We investigate the scalability of RoTS with respect to the number of expansion iterations and the scale of dataset size. As shown in Fig. 5(a), under the dataset size $100k$, increasing the number of expansion iterations from 0 to 32 improves the success rate from $15.8$ to $21.4$. This is because more iterations introduce a higher proportion of error-mode exploration and error-recovery trajectories into the dataset, enhancing the agent's reflection capability. In addition, we scale the dataset size from $50k$ to $1000k$ and report the corresponding performance in Fig. 5(b). The gains plateau at $1000k$ samples, achieving a success rate of $36.4$. We conjecture

that the performance saturates mainly due to current tree expansion setting: $N=4$ with 32 expansion iterations, which can not generate sufficiently diverse and effective trajectories. Therefore, further scaling the expansion parameters, *e.g.*, using a larger $N$ and more expansion rounds, may be beneficial.

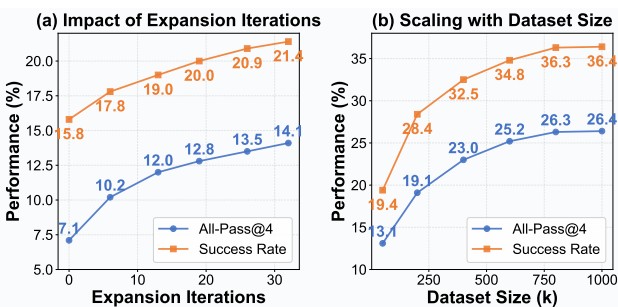

*Figure 5.* The scaling curve of RoTS with respect to the expansion round and dataset size.

**More Analysis.** We leave more detailed analysis of our work in Appendix F. We showcase an example of trajectory tree from our co-expansion in Fig 15, demonstrating the process of FDE and EIR. For example, failure cases of RoTS on OSWorld in Appendix F.5 show that RoTS occasionally demonstrates over-reflection and wastes inference budget. A cost analysis for data synthesis is provided in Appendix F.6. It shows that our method is cost-effective and easy to scale up.

## 5. Related Work

**Benchmarks for GUI Agents.** Existing GUI benchmarks predominantly evaluate grounding and perception accuracy (Cheng et al., 2024; Li et al., 2025), single-step accuracy conditioned on offline partial trajectories (Zheng et al., 2024; Li et al., 2024; Lu et al., 2025b), planning accuracy (Zheng et al., 2025) and overall task success rates in interactive environments (Bonatti et al., 2024; Xie et al., 2024; Rawles et al., 2025). Recent efforts further test robustness to environmental noise (Zhao et al., 2025; Yang et al., 2025b) adversarial attacks (Liao et al., 2025), and hallucination of agents (Zhang et al., 2025). However, existing benchmarks are dominated by hand-crafted perturbations, and are misaligned with the real distribution of policy-induced errors. Closest to our goal, AgentErrorBench (Zhu et al., 2025) studies failures of general LLM agents, whereas GUI agents operate in more complex multimodal environments with visually grounded actions and state changes. To fill this gap, we introduce GUI-RobustEval, a GUI benchmark covering diverse policy-induced error types, evaluating error awareness and post-error recovery from erroneous prefixes with controllable error depth.

**Data for Training Robust GUI Agents.** GUI agents are typically trained with supervised fine-tuning on trajectory data from videos, human demonstrations, or synthetic generation (Xu et al., 2025a;b; Wu et al., 2025c; Wang et al., 2025; Sun et al., 2025), but they still struggle with policy-induced errors in real-world execution (Wu et al., 2025a). Prior work improves reflection via offline reflection datasets (Wu et al., 2025a; Wanyan et al., 2025; Qin et al., 2025), online RL (Yang et al., 2025a; Lu et al., 2025a; Ye et al., 2025); yet offline data often over-represents short-horizon low-level mistakes, and online RL is constrained by sparse rewards and base-model limitations. Additionally, agent frameworks with reflection and backtrack modules (Wu et al., 2025b; Wang et al., 2024a; Agashe et al., 2025), as well as recent methods that enhance long-horizon execution via exploration-based data synthesis (Liu et al., 2025), compositional scheduling (Guo et al., 2025b; Deng et al., 2026), or persistent memory (Shi et al., 2026), can improve task success; however, they target general long-horizon capability rather than policy-induced error detection and recovery at the training level. Related to our goal, recent studies adopt a self-training paradigm to improve self-reflection (Zheng et al., 2025; Yuan et al., 2025). We instead propose an efficient tree-based sampling scheme that proactively explores diverse policy-induced errors and recovered trajectories. The resulting dataset can be used to improve the robustness to policy-induced errors of arbitrary GUI agents.

## 6. Conclusion

In this work, we find that current GUI agents are fragile to policy-induced errors, as existing training data under-covers the planning-level, long-horizon failures common in real execution. Motivated by this gap, we present GUI-RobustEval, a benchmark that quantitatively evaluates GUI agents' robustness to policy-induced errors. On the training side, we propose RoTS, a tree-based data synthesis framework that explores diverse error modes and generates reflection data on policy-induced errors. Experiments demonstrate that RoTS consistently improves robustness and overall performance, underscoring the value of long-horizon reflection for reliable GUI agents.

**Limitation.** We currently focus on desktop computer-use tasks; evaluating mobile and edge devices is left for future work. In GUI-RobustEval, evaluating from erroneous states requires injecting prefix histories into agents with heterogeneous formats, which inevitably involves cross-format conversion. While this conversion is applied consistently across all depths for a given agent, ensuring that within-agent degradation trends remain valid. We plan to use data flywheel or RL to iteratively improve both synthesis and model performance in a self-evolving manner.

## Impact Statement

This work improves the robustness of GUI agents, helping them detect and recover from their own mistakes rather than blindly continuing erroneous actions. We see this as a net positive for the safe deployment of autonomous computer-use systems. As computer-use agents play an increasingly important role in daily life and productivity, human judgment remains essential for high-stakes decisions and for verifying that agent behaviors align with user intent.

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

# A. More Details for GUI-RobustEval

## A.1. GUI-RobustEval Statistics

### A.1.1. DATA TYPES

We categorize agent errors into 11 types, as detailed in Table 6. Note that while the benchmark consists of 1,216 test cases across four error depths, the error type annotation is performed at the base trajectory level. We adopt multi-label because a single mistake can often be attributed to several root causes. The source trajectories are collected from 12 agents, including Jedi-7B (w/ o3 and GPT-4o), o3, Mobile-Agent-V3, GUI-Owl-7B, UI-TARS, OpenCUA (7B, 32B, and A3B), Kimi-VL-A3B, Doubao-1.5-Thinking, and AutoGLM.

*Table 6.* Error type distribution in GUI-RobustEval.

| Error Category | Description | Count | Avg. Success Rate |
|---|---|---|---|
| Incorrect UI Element | The agent interacted with the wrong UI element (*e.g.*, button, menu item) despite the correct option being available, often due to confusing semantically similar components. | 64 | 43.1% |
| Grounding Failure | The agent specified the correct action but failed to execute it accurately, such as clicking at imprecise coordinates, dragging incorrectly, or missing the interactive area. | 105 | 35.2% |
| Ineffective Action | The agent performs an action that results in no change to the environment state. | 68 | 42.6% |
| Typing Error | During typing, the agent produced incorrect text, usually accompanied by grounding errors that target the wrong editing range. | 21 | 45.4% |
| Miss Necessary Step | The agent skipped a critical action required for task completion, such as failing to click 'Save', or not pasting copied data. | 67 | 28.6% |
| Incorrect Tool Usage | The agent used an invalid, unsupported, or contextually inappropriate keyboard shortcut, terminal command, or function that cannot produce the intended effect. | 50 | 23.1% |
| Wrong Target | The agent operated on the incorrect file, cell range, column, slide, or data segment. | 40 | 12.7% |
| Incorrect Parameter | The agent entered or selected a wrong value, option (*e.g.*, font size, color). | 36 | 22.8% |
| Misunderstand Task Objective | The agent fundamentally misinterpreted the user's goal and pursued an unrelated objective. | 23 | 16.7% |
| Fail to Terminate | The agent fails to realize that the goal has already been achieved or is impossible to accomplish. | 13 | 11.1% |
| Lack of Knowledge | The agent selects an incorrect or inefficient strategy to reach the goal due to a lack of domain or application knowledge. | 31 | 22.9% |

### A.1.2. RELIABILITY OF ERROR-AWARENESS JUDGMENT

Error-Awareness is judged from the agent's first thought after takeover, where it may express the recognition of prior errors. We use Qwen3-VL-Plus as the judge model. To validate cross-agent reliability, we evaluate human agreement on 200 samples from GUI-RobustEval across three agents with different output styles (Table 7). The judge achieves ≥96% agreement across all agents, confirming robustness to output-style differences.

*Table 7.* Human agreement of the Error-Awareness judge across agents with different output styles.

| Agent | Agree. (%) | F1 |
|---|---|---|
| Qwen3-VL-Plus | 97.0 | 96.2 |
| UI-TARS-1.5 | 96.0 | 95.0 |
| OpenCUA-7B | 97.0 | 96.0 |

### A.1.3. DATA EXAMPLES

We provide four representative cases from GUI-RobustEval to illustrate diverse failure modes of SOTA GUI agent and the complexity of error recovery. The textual annotation beneath each screenshot describes the action the agent is prepared to execute in the current state. Actions verified as correct are marked in green, while errors are marked in red.

**Case 1: Incorrect Parameter.** In Fig. 6 (a), the task requires the agent to save CPU statistics to "System_Resources_Report.txt" on the Desktop. While the agent has the correct intent to monitor resources, it execute "sar -u -d 1 30  System_Resources_Report.txt" possibly because it incorrectly assumes the current directory is the Desktop. This Incorrect Parameter error results in the file being saved in the user home directory, failing the task requirement.

**Case 2: Miss Necessary Step.** Fig. 6 (b) showcases a Missed Necessary Step in LibreOffice Impress. The task is to export a slide as res.png. After opening the export dialog and selecting the PNG format (verified actions in green), the agent's next predicted action (in red) is to immediately click "Save" without entering the required filename. By skipping the naming step, the agent fails to produce the specifically requested file.

**Case 3: Incorrect UI Element.** Fig. 7 (a) illustrates Incorrect UI Element selection during text formatting in Word. After correctly selecting the target paragraphs, the agent attempts to set double line spacing, but repeatedly predicts click coordinates that target adjacent but incorrect buttons.

**Case 4: Compositional Error.** Fig. 7 (b) presents a complex compositional error, including Ineffective Action and Incorrect Tool Usage. The agent is tasked with installing a local VSIX file. Although it initially opens the Extensions view, it fails to recognize the state change. Consequently, it prepares to "re-click" the Extensions icon (an unnecessary action), but mistakenly opens the "File" menu instead. This error pollutes the agent's context, leading it to search for an "Install from VSIX" option in the wrong dropdown and eventually abandoning its original plan and open the VSIX file through the "File" menu.

(a) Task Instruction: Monitor Ubuntu system resource usage using the sar command from sysstat toolkit. Collect CPU statistics every second for 30 seconds and save the output to 'System_Resources_Report.txt' on Desktop.

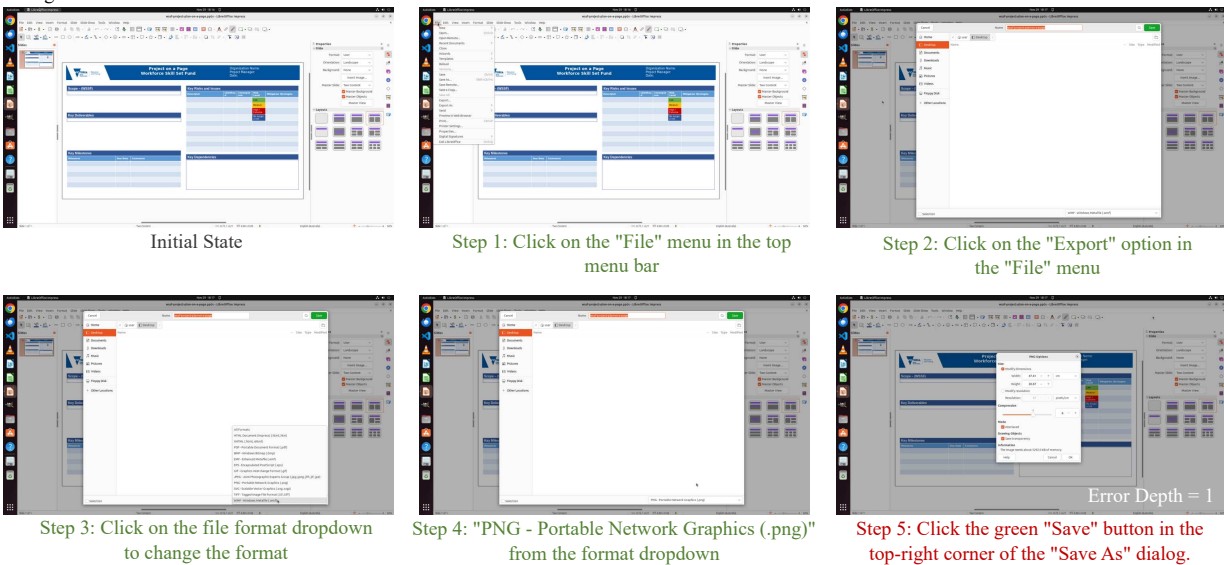

*Figure 6.* (a) Error example of Incorrect Parameter. The action description beneath each state indicates the agent's next move. In the final two steps, the agent fails to specify the correct output path in the terminal command. (b) Error example of Miss Necessary Step. The agent correctly navigates the export dialog but predicts an immediate save action (red text) before renaming the file to res.png.

## B. The Infrastructure

### B.1. Overview

The overview of our asynchronous online rollout system is shown in Fig. 8, which is used in both evaluation and data synthesis. Following OSWorld (Xie et al., 2024) and WindowsAgentArena (Bonatti et al., 2024), we host the Ubuntu and Windows systems on the Elastic Cloud Computing service, which achieves better parallelization than VMware and Docker. The GUI agents (*e.g.,* policy models, reward models and experience-informed reflector, *etc,*) are deployed as service in a distributed agent server that support parallel request from the rollout manager. The computation backend of these agents includes the self-hosted open-sourced GUI models and API-based proprietary models. The evaluation and data synthesis algorithm are implemented in the rollout manager that bridges environment server with agents server. Overall,

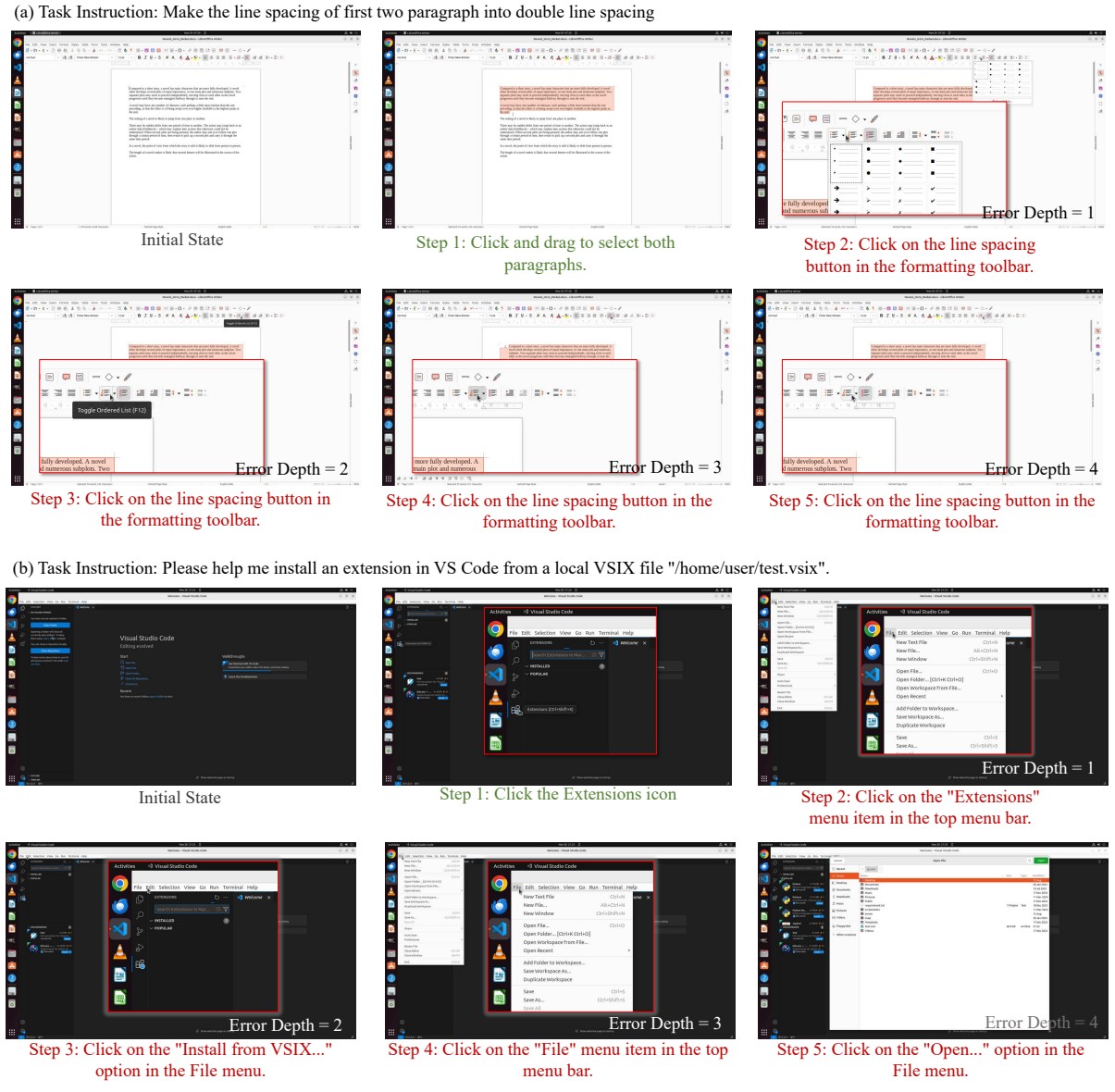

*Figure 7.* (a) Case study of Incorrect UI Element. The agent's predicted actions for the line spacing button target incorrect UI elements. (b) Case study of a Compositional Error. Initial perception failure leads to a chain of erroneous actions. The agent eventually loses track of the "Install extension" goal and attempts to open the VSIX as a regular file.

our infrastructure is flexible and efficient to (1) seamlessly scale to incorporate more agents and systems (*e.g.*, additional policy/reward/reflector variants and mobile-based operating systems), (2) support rapid extension to new rollout, evaluation, and data-synthesis algorithms by updating the rollout manager, and (3) enable high-throughput, high-concurrency sampling via asynchronous parallel rollouts across distributed environment and agent servers.

## B.2. Training Task Preparation

In this work, we adopt the same action space $\mathcal{A}$ as AgentNet. In total, $20k$ high-quality tasks are curated based on the following methods.

We select $10k$ tasks from AgentNet, $5k$ for Ubuntu and $5k$ for Windows respectively. First we install corresponding applications on the clean systems. Besides, we create corresponding accounts for the applications that require use-specific accounts, such as Thunderbird, YouTube, *etc.* Moreover, to achieve reliable environment replay, we close pop-ups for

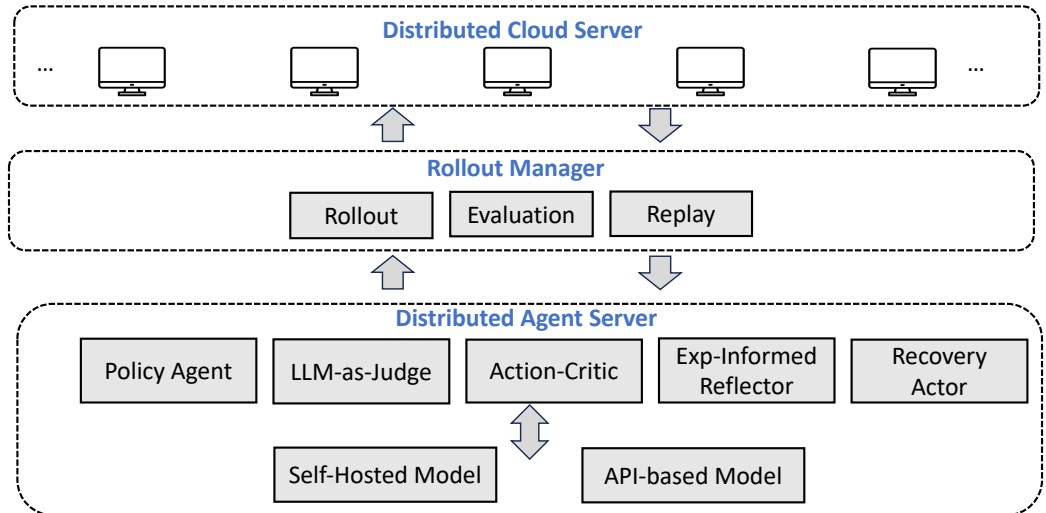

*Figure 8.* Overview of online sampling system for our GUI-RobustEval and RoTS.

browser, disable update notifications for both system and applications. The well-prepared Ubuntu and Windows systems are saved as the base snapshots for all the tasks. Second, we ask annotators to manually curate similar content (*e.g.,* documents, codes and pictures, *etc.*) used in each task and setup the base snapshot to the similar initial state as the AgentNet through a series of programmable functions (*e.g.,* uploading file to a specific path of the system, launching the corresponding application, *etc.*). These setup behaviors are recorded and saved in a configuration file for each task. To this end, we can achieve consistent and reproducible task initialization across different rollouts and tasks by applying recorded functions in the configuration file on the same base snapshot.

Apart from the $10k$ tasks from AgentNet, we further curate $10k$ high-quality, realistic tasks using the following procedure. Given a computer with various applications installed, inspired by PersonaHub (Ge et al., 2024), we first ask an LLM to role-play as individuals from different occupations and professional levels and to generate diverse everyday scenarios in which they use computers. Based on these scenarios, we then use LLMs to synthesize tasks. Finally, experienced annotators refine and verify the tasks, collect the necessary materials, and set the system to the corresponding initial states. These setup steps are recorded in a configuration file.

### B.3. Reward Model

#### B.3.1. A STUDY ON LLM-AS-JUDGE REWARD MODELS

In an online environment, we should also define the reward model $\mathcal{R}$, where $\mathcal{R}(\tau) \in \{0, 1\}$ takes the generated trajectories $\tau$ as input and outputs binary feedback to the agent: 1 for success and 0 for failure. In existing studies, state-checking (Xie et al., 2024) and LLM-as-Judge (Yang et al., 2025a) are two mainstream reward models for GUI tasks. The former verifies the final system state (*e.g.*, file existence, browser status, and system settings), while the latter leverages the ability of L(V)LMs or agents to evaluate the correctness of the trajectories. Since our dataset covers diverse tasks and checking the resulting system state for each task is infeasible, LLM-as-Judge is a more favorable approach.

Thus, we compare the performance of two common LLM-as-Judge methods, *i.e.*, ZeroGUI (Yang et al., 2025a) (a voting-based method) and WebJudge (Xue et al., 2025) (a three-stage framework). Specifically, we randomly sample 500 tasks from our task set and use Qwen3-VL-Plus as the policy model to perform rollouts. We use these two reward models to evaluate the correctness of the rollouts and ask expert annotators to double-check the evaluation results. We then compute human consistency and the F1 score based on whether the reward-model judgment aligns with human evaluation.

As shown in Table 8, WebJudge (Xue et al., 2025) achieves 90.00% accuracy on human consistency and has F1 Score of 77.42%, indicating a substantially better alignment with expert judgments. We attribute the advantage to its three-stage design, which explicitly decomposes the evaluation into structured steps and thus reduces ambiguity in interleaved image-text context. Therefore, we build our reward model for our online rollout system based on the framework of WebJudge.

*Table 8.* Comparison of different reward models.

| Reward Model | Human Consistency | F1 Score |
|---|---|---|
| ZeroGUI | 76.60% | 66.48% |
| **WebJudge** | **90.00%** | **77.27%** |

### B.3.2. WEBJUDGE FOR EXPERIENCE OUTPUTS

Compared to the original WebJudge, we adapt the judge to better support our experience-informed recovery setting: instead of only returning a scalar success/failure signal, our judge exposes reusable intermediate evaluation artifacts, including task-level procedures, step-wise state-transition summaries, and a procedure-aware diagnosis. These structured outputs can be directly consumed by downstream reflection and recovery modules.

Specifically, given a task instruction $u \in \mathcal{U}$ and the initial screenshot $o_1$, we first prompt a VLM to extract a set of task-level key procedures (milestones) $\mathcal{P}_u$ that characterize the expected progress for completing $u$. Given a trajectory $\tau = (o_1, a_1, o_2, \ldots, o_T)$, we then use another VLM to produce a step-wise summary of state transitions between consecutive screenshots, denoted as $\Delta_\tau = \{\delta_t\}_{t=1}^{T-1}$. Finally, an LLM with strong reasoning capability evaluates the trajectory by jointly conditioning on $(u, \mathcal{P}_u, \Delta_\tau)$, and outputs a binary reward $r_\tau \in \{0, 1\}$ together with a structured rationale $\xi_\tau$ that explains why the trajectory succeeds or fails, including the completion status of each procedure. We denote the judge output as:

$$(E_\tau, r_\tau) = \mathcal{R}(u, \tau), \qquad r_\tau \in \{0, 1\}, \tag{10}$$

where the *trajectory experience* is defined as

$$E_\tau \triangleq (\mathcal{P}_u, \Delta_\tau, \xi_\tau). \tag{11}$$

### B.3.3. PROGRESS CRITIC AND ACTION CRITIC

We also introduce a *progress critic* that evaluates whether a proposed low-level plan is feasible under the current UI state and whether it is likely to make local progress toward the task procedures. Following pre-operative critic (Wanyan et al., 2025), concretely, at each step $i$, given the current observation $o_i$ and an action by policy model $a_i$ together with the task instruction $u$ and history $h_{i-1}$, the critic outputs a binary score $c_i$ and a reasoning process $\zeta_i$:

$$(c_i, \zeta_i) = \mathcal{R}_p(u, o_i, a_i, h_{i-1}), \qquad c_i \in \{0, 1\}. \tag{12}$$

Additionally, we design a action accuracy critic model to evaluate step-level correctness of action:

$$\mathcal{R}_a(o_i, a_i, o_{i+1}) \in \{0, 1\}, \tag{13}$$

which is a VLM that takes two consecutive screenshots and an action as input, and outputs whether the action successfully leads to the expected result by comparing the two observations. $\mathcal{R}_p$ and $\mathcal{R}_a$ are necessary for predicting the fragile score and masking incorrect actions in the trajectory.

To validate the reliability of these critic modules, we randomly sample 200 steps from synthesized trajectories and compare critic predictions with expert human annotations. For $\mathcal{R}_p$, annotators judge whether the current action is a reasonable plan given the history and current observation; for $\mathcal{R}_a$, annotators additionally observe the next screenshot to verify execution correctness. As shown in Table 9, both critics achieve ≥88% agreement with human judgment, confirming that they provide reliable signals for downstream data filtering.

*Table 9.* Human agreement of critic modules.

| Module | Agreement (%) | F1 |
|---|---|---|
| Action critic $\mathcal{R}_a$ | 90.6 | 90.4 |
| Progress critic $\mathcal{R}_p$ | 88.7 | 88.7 |

We present the specific prompts for the reward models in Figures 9–13.

### B.3.4. REFLECTION IDENTIFIER $\mathcal{R}_f$

$\mathcal{R}_f$ is an LLM, which is designed for identifying reflection behaviors in the CoT within the trajectory of GUI agents. It takes input as the task instruction $u$, history $h_{i-1}$ to step $i$, prediction of current step, including the thought and action of current step $a_i$, and outputs the whether reflection behavior exists in $a_i$. Formally, it is denoted as:

$$\mathcal{R}_f(u, h_{i-1}, a_i) \in \{0, 1\}. \tag{14}$$

The prompt can be found in Fig. 14.

---

**Key-Point Extraction Prompt Template**

You are an expert in task planning. Based on the initial screen and the overall task goal, list the key sub-states or checkpoints required to complete the task. Respond with a numbered list. Each item should be a concise description of a state.
IMPORTANT: Describe only the states that need to be achieved. Do not include steps for verifying if a setting was successful.
TASK GOAL: "{instruction}"

---

*Figure 9.* Prompt template used for extracting key points (milestones) for reward modeling.

---

**State-Transition Summarization Prompt Template**

Below are two consecutive screenshots from a user's attempt to complete a task. The first image shows the state BEFORE an action, and the second image shows the state AFTER the action.
Please concisely describe the change or the action that occurred between these two screens.
**Input:** (Image 1) Screenshot before the action; (Image 2) Screenshot after the action.
**Output format:** A single concise sentence or short phrase describing the UI/state change.

---

*Figure 10.* Prompt template used to summarize state transitions between consecutive screenshots.

---

**Final Reward Judgment Prompt Template**

You are a meticulous evaluator. Your goal is to determine if a task was successfully completed by comparing the required steps with the actions taken, while applying sound judgment and contextual understanding.

**TASK GOAL:**

{instruction}

**REQUIRED KEY SUB-STATES (Checkpoints to achieve):**

{sub_states_str}

**ACTUAL ACTIONS TAKEN (Observed state changes):**

{transitions_str}

**EVALUATION:**

Based on all the information above, assess whether the agent successfully completed the task by reaching the final required state.

Please adhere to the following principles during your evaluation:

- **Distinction Between Action and Verification Sub-states:**

  - **Action Sub-states:** These are steps that involve performing a direct, necessary action (e.g., "delete the file," "type the text," "insert a table"). **All Action Sub-states MUST be completed for the task to be a success.** Failure to perform a required action is a failure of the task.
  - **Verification Sub-states:** These are steps designed to check or confirm the state of the system (e.g., "confirm the file is deleted," "check the text is correct"). A verification sub-state might be skipped if the agent achieves the outcome through a reliable alternative method (e.g., using a keyboard shortcut that does not provide visual feedback but still works).

- **Outcome is Paramount (in light of the above):** The ultimate criterion for success is whether the **task goal** was achieved. The task is a success if the final state is correct AND all **Action Sub-states** were completed. The sub-state list serves as a guideline, but the completion of action-oriented steps is non-negotiable.

- **On Saving Files:** Unless the task goal explicitly requires saving a file, the absence of a save action does not constitute a failure. The focus is on the state of the application or environment at the end of the task, not on the creation of an output file if it was not requested.

- **Flexibility in Method:** The agent may use an efficient or alternative approach that bypasses certain procedural steps (especially verification steps) but still fulfills the core objective. Do not penalize the agent for omitting non-critical verification steps if the final result is correct and achieved effectively.

Provide your reasoning clearly and thoroughly, then conclude with a single word on a new line: either `Success` or `Failure`.

*Figure 11.* Prompt template used for final task-success judgment in the reward model.

## C. More Details for RoTS Dataset

### C.1. Additional Details for Experience-Informed Recovery

C.1.1. TRAJECTORY EXPERIENCE FORMAT

Given instruction $u$ and trajectory $\tau = \{(o_t, a_t)\}_{t=1}^T$, the reward model outputs

$$(E_\tau, r_\tau) = \mathcal{R}(u, \tau), \qquad r_\tau \in \{0, 1\}, \tag{15}$$

where the trajectory experience is

$$E_\tau \triangleq (\mathcal{P}_u, \Delta_\tau, \xi_\tau). \tag{16}$$

$\mathcal{P}_u$ is a task-level procedure (milestone) list derived from $u$; $\Delta_\tau$ summarizes step-wise state transitions along $\tau$; and $\xi_\tau$ provides a diagnosis (reasoning trace) explaining why $\tau$ succeeds or fails.

---

**Progress Critic Prompt Template**

**System:** You are a Progress Critic for Computer Use. Before executing the proposed action $a_i$, judge if it is (1) feasible on the current screen $o_i$ and (2) likely to make progress toward the instruction $u$, given history $h_{i-1}$.

**Input:**
```
u:  {u}
h_{i-1}:  {h_{i-1}}
a_i:  {a_i}
o_i:  (screenshot image)
```

**Output:** Output ONLY JSON:

```
{
  "c_i": 0 or 1,
  "zeta_i": "<one-sentence reason;
            if reject, include the minimal fix/prerequisite>"
}
```

**Decision:** Set `c_i=1` only if the action target is visible and actionable on $o_i$, and the action is relevant to $u$ (not redundant/contradictory). If the target is missing/ambiguous, needs a prerequisite, or is risky without clear evidence from $u$ and $o_i$, set `c_i=0`.

---

*Figure 12.* Prompt template for the progress critic.

### C.1.2. NEIGHBORING-BRANCH TRAJECTORIES IN THE SEARCH TREE

Let the search tree contain nodes as observations/states and directed edges as actions. For a node $o$, denote by $\text{Out}(o)$ the set of outgoing edges:

$$\text{Out}(o) \triangleq \{(o, a, o') \mid \text{taking action } a \text{ at } o \text{ leads to child node } o'\}. \tag{17}$$

Consider a failed trajectory $\tau^{\text{fail}} = \{(o_i, a_i^{\text{fail}})\}_{i=1}^T$ (root node at $i = 1$). For any step $i > 1$, we define the *neighboring branches* at prefix node $o_i$ as all outgoing edges that take actions different from the failed one:

$$\mathcal{B}(o_i) \triangleq \{(o_i, a, o') \in \text{Out}(o_i) \mid a \neq a_i^{\text{fail}}\}. \tag{18}$$

Each branch edge $(o_i, a, o') \in \mathcal{B}(o_i)$ induces a set of full trajectories by following any continuation in the subtree rooted at $o'$. We write this mapping as

$$\text{Traj}(\mathcal{S}) \triangleq \bigcup_{(o,a,o') \in \mathcal{S}} \{\text{all complete trajectories starting with edge } (o, a, o')\}. \tag{19}$$

The set of neighboring trajectories of $\tau^{\text{fail}}$ is then

$$\mathcal{N}(\tau^{\text{fail}}) \triangleq \bigcup_{i=2}^T \text{Traj}\big(\mathcal{B}(o_i)\big). \tag{20}$$

We construct the corresponding *neighbor experience set* as

$$\mathcal{E}(\tau^{\text{fail}}) \triangleq \{E_{\tau^{\text{nb}}} \mid \tau^{\text{nb}} \in \mathcal{N}(\tau^{\text{fail}})\}. \tag{21}$$

### C.1.3. REFLECTION WITH NEIGHBORING-BRANCH EXPERIENCE

Given a failed trajectory $\tau^{\text{fail}}$, we collect experiences from its neighboring (sibling-branch) trajectories into a neighbor-experience set $\mathcal{E}(\tau^{\text{fail}})$. Conditioning on $(u, \tau^{\text{fail}}, \mathcal{E}(\tau^{\text{fail}}))$, the experience-informed reflector produces $K$ candidates:

$$\{(i_k, g_{i_k}, p_{i_k})\}_{k=1}^K \sim \pi_\theta^{er}\big(u, \tau^{\text{fail}}, \mathcal{E}(\tau^{\text{fail}})\big), \tag{22}$$

---

**Step-Level Action Critic Prompt Template (with Transition Context)**

You are given two consecutive screenshots from a GUI trajectory. The first image is the state **before** the action, and the second image is the state **after** the action.
**State-transition context (summary of the observed change):**
{transition_description}
**Intended action:**
{action}
Based on the visual evidence and the transition context, decide whether the intended action was successfully executed. If the observed change matches the intended action, respond with `success`; otherwise, respond with `fail`.
Respond with the following format:
**Thought:** "your thought"
**Conclusion:** "success/fail"

---

*Figure 13.* Prompt template for the step-level action critic that verifies whether an intended action is consistent with the observed state transition. The transition state is obtained by the state-transition summarization prompt for the reward model.

where $i_k$ is a proposed error step index, $g_{i_k}$ is the corresponding recovery advice, and $p_{i_k} \in [0, 1]$ is the expansion priority.

To balance exploiting high-priority candidates and exploring less-visited recovery points, we maintain a recovery visit count $V_i^r$ for each step/node and select the recovery point with a UCB-style score:

$$s_{i_k} = p_{i_k} + c\sqrt{\frac{\ln\left(V_{p(i_k)}^r + 1\right)}{V_{i_k}^r + 1}}, \qquad k^* = \arg\max_{k \in \{1,\ldots,K\}} s_{i_k}, \qquad i^* = i_{k^*}. \tag{23}$$

### C.1.4. ADVICE-CONDITIONED RECOVERY ROLLOUT

After selecting $i^*$, we replay the environment to the corresponding prefix node $o_{i^*}$ (and the prefix history $h_{i^*-1}$ if applicable). A recovery actor then generates a rollout conditioned on the advice:

$$\tau^{\text{rec}} \sim \pi_\theta^{rec}(u, o_{i^*}, h_{i^*-1}, g_{i^*}). \tag{24}$$

Equivalently, if the failed trajectory is $\tau^{\text{fail}} = \left((o_1, a_1^{\text{fail}}), \ldots, (o_T, a_T^{\text{fail}})\right)$, then the recovered trajectory can be written as

$$\tau^{\text{rec}} = \underbrace{\left((o_1, a_1^{\text{fail}}), \ldots, (o_{i^*-1}, a_{i^*-1}^{\text{fail}})\right)}_{\text{replayed prefix}} \| \underbrace{\left((o_{i^*}, a_{i^*}^{\text{rec}}), \ldots, (o_{i^*+L}, a_{i^*+L}^{\text{rec}})\right)}_{\text{actor-sampled suffix}}, \tag{25}$$

where $L \geq 0$ is the rollout length.

### C.2. Case Study of FAR-Tree Expansion

To intuitively illustrate the dynamics of our data synthesis framework, we provide a case study of a representative FAR-Tree in Fig. 15. The tree is constructed through 4 initial parallel samplings followed by 8 rounds of joint FDE and EIR expansions.

The **orange dashed region** depicts a scenario where a successful trajectory undergoes a policy-induced deviation triggered by FDE, which is subsequently rectified via EIR. Specifically, at node `58cfbf4a`, the task is effectively complete, and the optimal action is termination. However, under FDE perturbation, the agent performs a redundant and harmful operation by removing "English" from the configured languages—an action that contradicts the user instructions. By node `483ce135` (two steps after the deviation), the "English" option has been removed. At this failure point, the EIR module generates the following guidance: *"You should restore English, as the goal does not specify its removal."* The advice-conditioned recovery actor follows this instruction, re-inserts "English," and ultimately achieves a successful task completion.

The **blue dashed region** highlights the correction of a planning error originated from parallel sampling. In this instance, the agent initially adopts an inefficient scrolling strategy to locate a target language (nodes `b2c64a22` to `cffa6fbd`). The EIR reflector identifies this sub-optimal behavior at node `cffa6fbd` and suggests: *"Stop scrolling; you should search for 'Japanese' directly in the search bar."* Guided by this experience-informed advice, the recovery agent immediately activates the search bar and completes the task efficiently.

---

**Reflection-on-Erroneous-Behavior Evaluator Prompt**

**You are a strict evaluator. Your only job is to decide whether there is any "reflection on erroneous behavior".**

**You will be given:**

- The overall task goal: instruction (what needs to be accomplished)

- Historical `action_instructions` (excluding the current node)

- The current thought (the model's self-report / reasoning)

**Decision rules (very important):**
Only when the thought explicitly contains **BOTH** of the following do we count it as "reflection":

1. It states that a prior action/decision/step was wrong relative to the instruction (i.e., it was incorrect, off-track, failed to meet requirements, or was a bad choice). It must be an explicit judgment of error / non-compliance with the task goal.

2. It includes an intention to correct / an improvement strategy (e.g., "therefore I need to change to...", "roll back...", "reselect...", etc.).

If the thought only contains any of the following, it must be judged as **NO reflection** (`has_reflection=false`):

- "The previous step succeeded / completed / now I will do the next step" (success assessment)

- "The previous step was extra/redundant but does not affect correctness" (pure redundancy without pointing to an error)

- Merely describing the UI or planning the next step, without explicitly stating "it was wrong and needs correction"

**Output must be JSON only (no extra text), with the format:**

```
{
  "has_reflection": true/false,
  "reflection_type": "error_analysis|strategy_fix|none",
  "error_target": "What kind of error is being reflected on
      (e.g., wrong scope / wrong chart type / missing legend placement /
      forgot to save). Empty string if none.",
  "evidence": "The most critical original phrase excerpted from the thought",
  "confidence": 0.0-1.0
}
```

*Figure 14.* Prompt used for the reflection identifier.

## C.3. CoT Synthesis Procedures

Our trajectories are collected from multiple policy models that differ in action spaces and CoT styles. To enable joint training, we unify all trajectories to the AgentNet format (Wang et al., 2025). For each source policy, we define an action mapping from its native action space to AgentNet's action space and normalize coordinate-based actions (*e.g.,* click, drag, right click) to a $(0, 1)$.

We further rewrite intermediate thoughts into a unified CoT/reflection format. Specifically, we adapt the AgentNet CoT generator and instantiate it with Qwen3-VL-Plus. The generator conditions on the task instruction, trajectory history, current screenshot, the original thought from the policy model, and the reflection signals produced by our critics (*i.e.*, critic-generated thoughts). It outputs a rewritten CoT/reflection trace, which is concatenated with the canonicalized executable action to form the step target sequence. In our training data, we also deploy the same system prompt as AgentNet (Fig. 16).

Task Instruction: please help me add Japanese and Russian to the preferred language of Chrome browser.

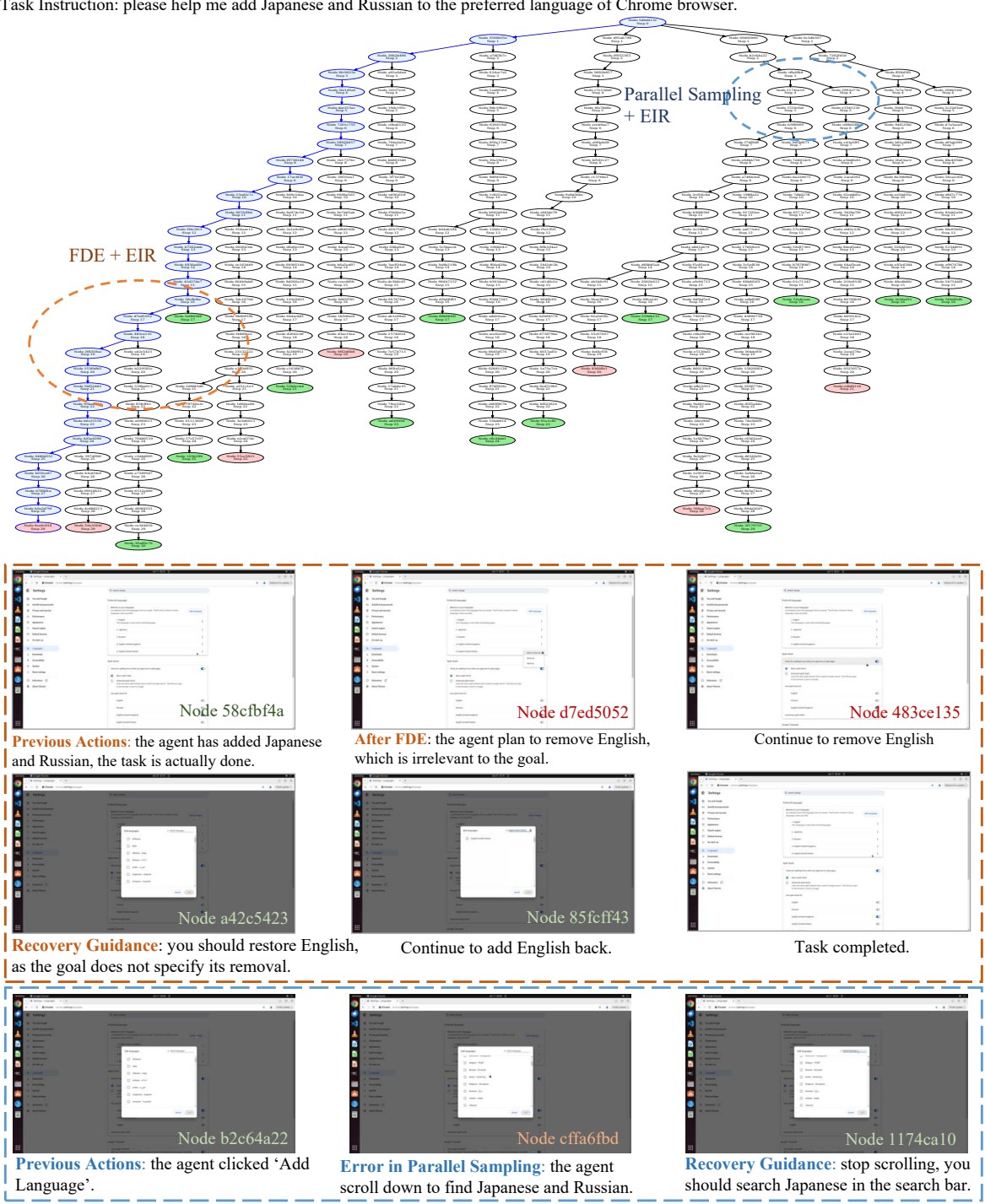

*Figure 15.* Visualization of the FAR-Tree, illustrating policy-induced errors via parallel sampling and FDE and their subsequent recovery through EIR advice.

## C.4. Rule-Based Data Deduplication

After obtaining $D_{\text{agn}}$ and $D_{\text{ref}}$, we design the following rule-based data deduplication methods to remove near-duplicate data to preserve diversity. Specifically, we balance the number of training instances across tasks. Within each task, we tokenize the concatenation of `thought` and `action` and form a set of token $n$-grams, and use MinHash to obtain a compact signature for this set. Two instances are treated as duplicates if the similarity estimated from their MinHash signatures

---

**System Prompt for Training**

You are a GUI agent. You are given a task and a screenshot of the screen. You need to perform a series of PyAuto-GUI actions to complete the task. The password of the computer is `"password"`. If the task is not possible, output `computer.terminate(status='failure')`.

For each step, respond in the following format:

**Thought:**

- **Step-by-Step Progress Assessment:**

  - Analyze completed task parts and their contribution to the overall goal.
  - Reflect on potential errors, unexpected results, or obstacles.
  - If the previous action was incorrect, predict a logical recovery step.

- **Next Action Analysis:**

  - List possible next actions based on the current state.
  - Evaluate options considering the current state and previous actions.
  - Propose the most logical next action.
  - Anticipate consequences of the proposed action.

- **For Text Input Actions:**

  - Note the current cursor position.
  - Consolidate repetitive actions (specify count for multiple keypresses).
  - Describe the expected final text outcome.

- Use first-person perspective in reasoning.

**Action:** Provide clear, concise, and actionable instructions.

- If the action involves interacting with a specific target:

  - Describe the target explicitly without using coordinates.
  - Specify element names when possible (use the original language if non-English).
  - If the name is unavailable, describe distinctive features (shape, color, and position).
  - For window control buttons, identify correctly: minimize ``--'', maximize ``□'', close ``X''.

- If the action involves keyboard actions (*e.g.*, `press`, `write`, `hotkey`):

  - Consolidate repetitive keypresses with a count.
  - For typing actions, specify the expected text outcome.

Finally, output the action as PyAutoGUI code, or the following function:

```
{"name": "computer.terminate",
 "description": "Terminate the current task and report its completion status",
 "parameters": {"type": "object",
                "properties": {"status": {"type": "string",
                                          "enum": ["success","failure"],
                                          "description": "The status of the task"}},
                "required": ["status"]}}
```

*Figure 16.* System Prompt template used for training our agent.

exceeds a threshold and we keep a single representative among duplicates.

## C.5. Data Samples from RoTS Dataset

In this section, we show some training samples from our dataset.

### C.5.1. SELF-REFLECTION DATA EXAMPLE FOR SHORT ERROR

```
1  [
2      {
3          "role": "system",
4          "content": system_prompt_for_training
5      },
6      {
7          "role": "assistant",
8          "content": "# Step 1:
9          ## Action: Click the "Colors" menu in GIMP's top menu bar to open its dropdown."
10
11         # Step 2:
12         ## Action: Click the "Hue-Saturation..." option in the open Colors menu to open the Hue-
                  Saturation adjustment dialog."
13     },
14     {
15         "role": "user",
16         " image " :
```

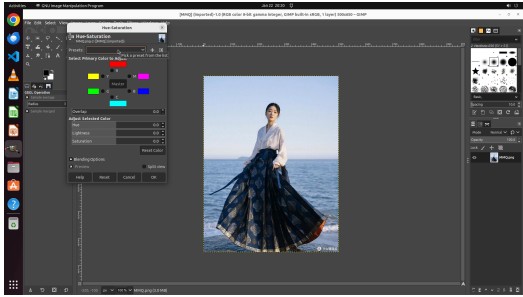

```
1      },
2      {
3          "role": "assistant",
4          "content": "# Step 3:
5          ## Action: Click the Hue numeric entry box (showing "0.0") in the Hue-Saturation dialog to focus
                  the Hue value field."
6      },
7      {
8          "role": "user",
9          " image " :
```

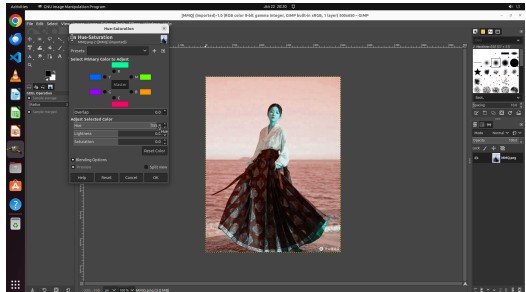

```
1        },
2        {
3            "role": "assistant",
4            "content": "# Step 4:
5            ## Action: Type "5.5" into the currently focused Hue numeric input field to set the Hue value to
                   5.5."
6        },
7        {
8            "role": "user",
9            " image " :
```

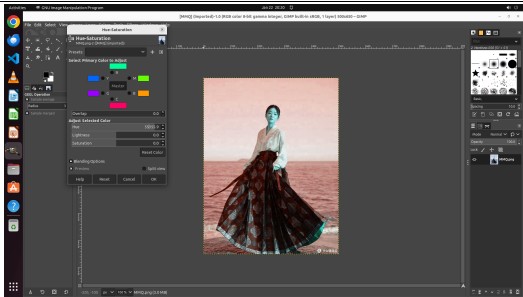

```
1        },
2        {   "role": "user",
3            "content": "# Task Instruction:
4              Could you please make the colors of the current picture more vivid through the Hue-Saturation
                       function in GIMP? Set the hue to 5.5, the lightness to 6.7, and the saturation to 9.8.
5              Please generate the next move according to the screenshot , task instruction and previous steps
                   (if provided)."
6        }
7        {
8            "role": "assistant",
9            "content": "# Step 5:
10
11           ## Thought: Previously, typing "5.5" into the Hue field did not replace the existing value;
                   instead it appended to the current number, leaving "55155.9". This suggests the text was
                   inserted without selecting the existing content, so the field now contains an incorrect, very
                    large value.
12           The dialog is open and responsive, and the Hue field is focused. To correct this, I should select
                   all text in the Hue input so I can overwrite it with the exact value "5.5". After fixing Hue
                   , I will proceed to set Lightness to 6.7 and Saturation to 9.8.
13            The next logical step is to select the entire contents of the Hue numeric box so the next
                   keystrokes replace it rather than append. Pressing Ctrl+A while the input is focused will
                   highlight the full value, preparing it for replacement.
14
15           ## Action: Press Ctrl+A to select all text in the currently focused Hue numeric input field so
                   the entire value is highlighted for replacement.
16
17           ## Code: '''pyautogui.hotkey('ctrl', 'a')'''
18            "
19        }
20    ]
```

## C.5.2. SELF-REFLECTION DATA EXAMPLE FOR LONG ERROR

```
1  [
2      {
3          "role": "system",
4          "content": system_prompt_for_training
5      },
6      {
7          "role": "assistant",
8          "content": "# Step 1:
9      ## Action: Click the Terminal icon in the left dock (the dark square with a white command prompt
           symbol) to open a terminal window."
10
11         # Step 2:
12     ## Action: Click the Terminal icon in the left dock (the dark square with a white command prompt
           symbol) to bring the terminal to the foreground."
13     },
14     {
15         "role": "user",
16         " image " :
```

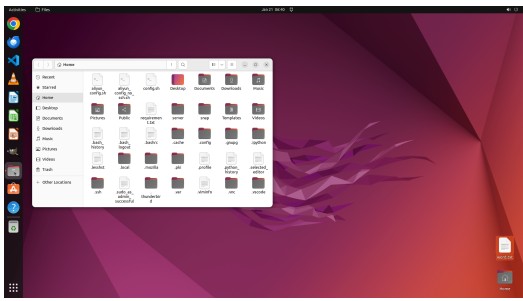

```
1      },
2      {
3          "role": "assistant",
4          "content": "# Step 3:
5      ## Action: Type exactly: cd ./Desktop\n This enters the command to change into the Desktop
           directory and executes it."
6      },
7      {
8          "role": "user",
9          " image " :
```

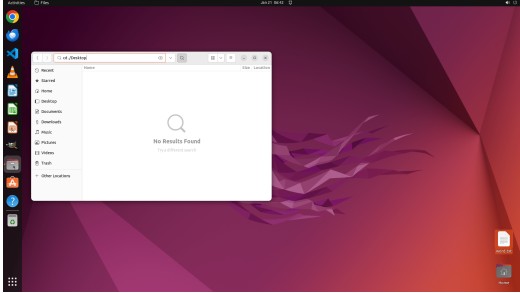

```
1        },
2        {
3            "role": "assistant",
4            "content": "# Step 4:
5            ## Action: Type exactly: last | grep shutdown > word\n This enters the command to run "last",
                filter lines containing "shutdown", and redirect the output to a file named "word" in the
                current directory, overwriting the file if it exists."
6        },
7        {
8            "role": "user",
9            " image " :
```

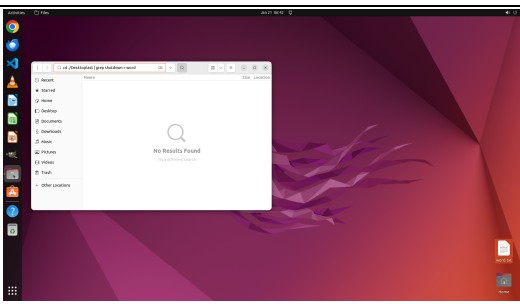

```
1        },
2        {   "role": "user",
3            "content": "# Task Instruction:
4             Open the terminal, use the command \"w\" to check which users are currently logged in, then
                enter the directory \"./Desktop\", use the \"last\" command to query the system's last
                restart time and output the result to a file named \"word\" (overwriting the original
                content), and then view the first 5 lines of the file \"word\"."
5        }
6        {
7            "role": "assistant",
8            "content": "# Step 5:
9
10           ## Thought: I previously typed the entire sequence of commands into the Files search box instead
                of a terminal, which is why the search shows "No Results Found." To correct this, I need to
                get back to a terminal where I can execute the commands. The terminal is already open but not
                focused; the Files window is in front.\n\nThe next step is to close the Files window so I
                can access the terminal behind it. Clicking the close button ("X") on the Files window will
                dismiss it, revealing the terminal. Once the terminal is visible, I can type the correct
                commands: run "w", then change to ./Desktop, run "last > word", and finally "head -5 word".\n
                \nThe most logical action now is to close the Files window by clicking its close button at
                the top-right of the window. After this, the terminal should come to the foreground, allowing
                me to proceed with entering the commands.
11
12           ## Action: Click the close button ("X") at the top-right corner of the Files window to close it.
13
14           ## Code: '''pyautogui.click(968, 211)'''
15            "
16       }
17  ]
```

# D. Benchmarks and Baselines

## D.1. Benchmarks

### D.1.1. Setup for GUI-RobustEval

The GUI-RobustEval contains 1216 test cases and 11 typical types of errors of GUI agents. The error depth $d$ varies from 0 to 5, where 0 refers to the verified-correct prefix. And the maximum number of steps that the agent can perform is 50, which includes the erroneous partial prefix. The Error-Awareness Rate and Post-Error Success Rate with respect to different error depth and error-types are reported by averaging 3 independent runs. Benefit from the same infrastructure as in our data synthesis pipeline, our evaluation is highly parallel, running 50 tasks at the same time.

### D.1.2. Setup for End-to-End Benchmarks

We use OSWorld-Verified and WindowsAgentArena to evaluate the performance of RoTS on end-to-end tasks. OSWorld-Verified contains a Ubuntu Desktop environment and 369 tasks covering diverse applications in open domain. Each task is configured with a task instruction, setup file for the initial state, and a rule-based evaluator. While WindowsAgentArena (Bonatti et al., 2024) has 154 tasks running on Windows 11 system. It has a similar framework to OSWorld and reflects the agent's performance on Windows-centric applications. Similarly, we host the environment of these two benchmarks on ECS to achieve better parallelization, running 50 tasks in parallel. We report our success rate under different step budgets (15 and 50). In addition, to quantify robustness, we report All-Pass@4, *i.e.*, the fraction of tasks solved in all four independent runs.

## D.2. Baseline Models

### D.2.1. Baseline Methods on GUI-RobustEval

We compare RoTS with representative GUI agents from both proprietary and open-source families, covering agentic models and planner-grounder frameworks. For proprietary models, we include Claude4.5-Sonnet (Anthropic, 2025), Doubao1.5 (Guo et al., 2025a), Qwen3-VL-Flash/Plus (Bai et al., 2025a), and several vision-language GUI-oriented models (*e.g.*, UI-TARS1.5 (Qin et al., 2025)), all accessed via their official APIs. For open-source baselines, we evaluate GUI-Owl-7B (Ye et al., 2025), Qwen3VL-8B-Instruct (Bai et al., 2025a), UI-TARS1.5-7B (Qin et al., 2025), and OpenCUA (7B/32B) (Wang et al., 2025), which cover general-purpose VLM instruction-tuned models and GUI-specialized agents. The open-sourced agents and our models are deployed on a server with 32 NVIDIA A100 GPUs. For planner-grounder architectures, we evaluate the open-weight Jedi-7B (Xie et al., 2025) with GPT-o3 (OpenAI, 2025a) as the planner. We additionally report our RoTS models trained with RoTS-7B and RoTS-32B under the same evaluation protocol. The temperature of tested agents is set as the reported value if explicitly specified otherwise 0.

### D.2.2. Baseline Methods on OSWorld

We compare RoTS against both proprietary and open-weights GUI agents on OSWorld. Proprietary baselines include OpenAI CUA (OpenAI, 2025b) and UI-TARS-1.5 (Qin et al., 2025) (agentic GUI models), as well as strong general-purpose multimodal LLMs (Claude 3.7/4.5 Sonnet (Anthropic, 2025), Doubao-1.5-Thinking (Guo et al., 2025a), and Qwen3-VL-Flash/Plus (Bai et al., 2025a)) used as GUI agents under the OSWorld protocol. Open-weights baselines cover established agentic models (OpenCUA (Wang et al., 2025), UI-TARS-1.5-7B (Qin et al., 2025) and GUI-OWL (Ye et al., 2025)), and general-purpose VLMs (Qwen2.5-VL (Bai et al., 2025b), Qwen3-VL Thinking (Bai et al., 2025a)) across multiple model sizes.

The open-sourced agents and our models are deployed on a server with 32 NVIDIA A100 GPUs, while the proprietary models are called through APIs from the corresponding provider. The temperature of tested agents is set as the reported value if explicitly specified otherwise 0.

### D.2.3. Baseline Methods on WindowsAgentArena

We compare RoTS with representative GUI agents and strong multimodal LLMs on WindowsAgentArena, including Claude 3.7 Sonnet (Anthropic, 2025) and Qwen2.5-VL-72B (Bai et al., 2025b) as general-purpose VLM baselines; UI-TARS (7B and 72B-DPO) (Qin et al., 2025) as GUI-specialized agentic models; and open weights CUA agents (OpenCUA (Wang et al., 2025) and ScaleCUA (Liu et al., 2026)). We also include Jedi-7B w/ GPT-4o (Xie et al., 2025), a hybrid agent that pairs an open-weights policy with a proprietary planner.

# E. Implementation Details

## E.1. Dataset Synthesis

We construct $20k$ online tasks (Appendix B.2). To improve dataset diversity and generalization, we adopt three policy models: UI-TARS-1.5-7B, OpenCUA-7B, and Qwen3-VL-Plus. UI-TARS-1.5-7B and OpenCUA-7B are two open-weights GUI specific agents, which are deployed on a server with 32 NVIDIA A100 GPUs, while Qwen3-VL-Plus is accessed via API. We set the sampling temperature to 0.1, while the maximum rollout step is set as 30. The desktop environment is hosted on ECS with distributed serving, enabling up to 120 tasks to run in parallel.

For tree expansion, we use $N = 4$ parallel rollouts to build the initial tree for each policy model, followed by 32 co-expansion rounds. This yields 68 trajectories per task for each policy model. The exploration-exploitation trade-off $c$ for UCB in Eq. 1 and Eq. 5 is set as 0.25, which follows the implementation in Agent-R (Yuan et al., 2025). We use Qwen3-VL-Plus as the backbone model for experience-informed reflector $\pi^{er}$ and recovery actor $\pi^{rec}$ to perform error state identification and error recovery.

We implement the reward function $\mathcal{R}$ with three components. We use Qwen3-VL-Plus for keypoint extraction, state-transition modeling. We use Qwen-Max for final reward judgment. The action-critic $\mathcal{R}_a$ takes state-transition as input, which is pure text. Thus, Qwen-Max can be used. The progress critic model and the reflective behavior identifier $\mathcal{R}_p$ and $\mathcal{R}_f$ are implemented with Qwen3-VL-Plus.

We use the progress critic and action critic model to remove incorrect steps in the trajectory and split the remaining steps into reflection-agnostic and reflection-related subsets with $\mathcal{R}_f$. After the MinHash du-duplication, we obtain the final $\mathcal{D}_{\mathrm{agn}}$ and $\mathcal{D}_{\mathrm{ref}}$. Table 10 reports detailed statistics with respect to tree-based expansion and the final resulting dataset. In this table, we report the accuracy rate of these policy models when at least 1 correct trajectory exists across 68 rollouts and the average success rate. Additionally, the reflection-agnostic and reflection-related dataset size after post-processing is reported.

*Table 10.* Dataset statistics after tree expansion by three policy models. Pass@68 denotes the percentage of tasks for which at least one of the 68 sampled trajectories succeeds after tree expansion, while the average is the averaged success rate calculated by the ratio of correct trajectories to the total number of trajectories. $|\mathcal{D}_{\mathrm{agn}}|$ and $|\mathcal{D}_{\mathrm{ref}}|$ represent the number of trainable steps of reflection-agnostic and reflection-related datasets.

| Policy Model | Pass@68 (%) | Average (%) | $|\mathcal{D}_{\mathbf{agn}}|$ | $|\mathcal{D}_{\mathbf{ref}}|$ |
|---|---|---|---|---|
| UI-TARS-1.5-7B | 48.2 | 20.5 | $400k$ | $150k$ |
| OpenCUA-7B | 50.4 | 22.4 | $510k$ | $200k$ |
| Qwen3-VL-Plus | 55.3 | 25.6 | $600k$ | $350k$ |
| **Overall** | 61.2 | 22.8 | $1510k$ | $700k$ |

## E.2. Training Details

We fine-tune QWEN2.5-VL-7B for one epoch on 64 NVIDIA A100 GPUs and QWEN2.5-32B for one epoch on 128 NVIDIA A100 GPUs; both are trained with a global batch size of 512. For both models, we use DeepSpeed ZeRO-3 with gradient checkpointing, bfloat16 precision, and FlashAttention, and optimize with AdamW using a learning rate of $1 \times 10^{-5}$ and a warmup ratio of 0.05. The maximum sequence length is 32,768 tokens. For both settings, we set the maximum number of image history to 5.

# F. More Analysis

RoTS-7B and RoTS-32B achieve competitive success rate on WindowAgentArena, which attain 28.2 and 39.1 at max step 50, which surpassing previous open-weights GUI agents and agent frameworks such as Jedi-7B w/ GPT4-o.

## F.1. Results on WindowsAgentArena

*Table 11.* Comparison of the state-of-the-art methods on the WindowsAgentArena benchmark. We report the success rate (%) as the evaluation metric. † denotes our reproduced results, averaged across 4 independent runs. The best and second best performing models are marked by bold.

| Agent Method | Success Rate (%) | |
|---|---|---|
| | Max Steps: 15 | Max Steps: 50 |
| Claude 3.7 Sonnet(Anthropic, 2025) | 7.1 | 6.4 |
| Qwen2.5-VL-72B(Bai et al., 2025b) | 11.8 | 9.7 |
| Jedi-7B w/ GPT4-o(Xie et al., 2025) | 30.2 | 32.9 |
| UI-TARS-1.5-7B(Qin et al., 2025) | 11.1 | 15.9 |
| UI-TARS-72B-DPO(Qin et al., 2025) | 11.1 | 17.9 |
| OpenCUA-7B(Wang et al., 2025) | 13.5 | - |
| ScaleCUA-7B(Liu et al., 2026) | 18.0 | 20.7 |
| ScaleCUA-32B(Liu et al., 2026) | 21.4 | 24.2 |
| **RoTS-7B** | **24.9**† | **28.2**† |
| **RoTS-32B** | **35.9**† | **39.1**† |

## F.2. Per-Error-Type Analysis on GUI-RobustEval

To provide fine-grained diagnostics, we report the per-error-type post-error success rate of RoTS-32B on GUI-RobustEval in Table 12, averaged across all depths. The baseline column corresponds to the average post-error success rate reported in Table 6.

*Table 12.* Per-error-type post-error success rate (%) on GUI-RobustEval, averaged across depths. Baseline Avg. corresponds to the Avg. Success Rate in Table 6.

| Error Type | Baseline Avg. | RoTS-32B | Improv. |
|---|---|---|---|
| Incorrect UI Element | 43.1 | 52.3 | +9.2 |
| Grounding Failure | 35.2 | 42.7 | +7.5 |
| Ineffective Action | 42.6 | 53.9 | +11.3 |
| Typing Error | 45.4 | 54.8 | +9.4 |
| Miss Necessary Step | 28.6 | 38.2 | +9.6 |
| Incorrect Tool Usage | 23.1 | 30.8 | +7.7 |
| Wrong Target | 12.7 | 20.9 | +8.2 |
| Incorrect Parameter | 22.8 | 31.4 | +8.6 |
| Misunderstand Task Objective | 16.7 | 24.1 | +7.4 |
| Fail to Terminate | 11.1 | 17.4 | +6.3 |
| Lack of Knowledge | 22.9 | 33.4 | +10.5 |

RoTS-32B brings broad improvements across all error types. The largest gain appears on *Ineffective Action* (+11.3), where richer exploration and experience-informed recovery provide the most benefit. Bottlenecks remain in *Fail to Terminate* (17.4%) and *Misunderstand Task Objective* (24.1%), where recovery requires accurate progress perception and re-planning over long histories—important directions for future work.

## F.3. Trajectory Comparison of Existing GUI Agent and RoTS Model on GUI-RobustEval

In Fig. 17, we present a case study where the agent is tasked with disabling a specific website from opening automatically at startup. In the replayed trajectory Fig. 17 (a), the agent initially succeeds in setting the startup behavior to "Open the New Tab page" but subsequently commits a "Fail-to-Terminate" error, drifting into task-irrelevant and harmful operations. The

baseline model, OpenCUA, continues to execute harmful actions such as removing "Bing" from the search engine list as shown in Fig. 17 (b). Our model successfully identifies the task deviation. It navigates back to the "On start-up" settings page and verify that the target website has indeed been removed as shown in Fig. 17 (c).

### F.4. Exploration and Error Recovery Behavior of RoTS Model on OSWorld

In Fig. 18, we demonstrate an example of how our model successfully completes a task on OSWorld through flexible exploration strategy switching and several error recovery steps. The task requires the agent to "Find the FAQ page about ticket delivery." When the initial page indicates that the website is inaccessible, the agent first attempts to go back to the previous page and then tries re-entering the URL; however, both exploration attempts fail to resolve the issue. Consequently, the agent shifts its strategy to direct searching but encounters an execution issue (a Typing Error). Upon identifying the error, the agent performs a recovery by using Ctrl+A to select the existing text and re-typing the correct keywords, ultimately fulfilling the task requirements.

### F.5. Failure Cases of RoTS Model on OSWorld

Fig. 19 illustrates an instance of "over-reflection" by our model during a task on OSWorld. The task requires the agent to "Show me all men's large-size short-sleeve shirts with a discount of 50% or more." During the execution, the webpage fails to render completely. Misinterpreting this incomplete state as an incorrect page, the agent mistakenly performs two consecutive "go back" operations. In reality, the task could have been successfully advanced from the current page; however, the agent's excessive self-correction led to these redundant and unnecessary back-navigation steps.

### F.6. Cost Analysis

We summarize the cost and wall-clock time for our benchmark construction (1216 test cases) and data synthesis (20K tasks). The overall expense comes from three components: human effort, policy/model inference, and environment runtime.

For benchmark construction, it involves human effort for task filtering and verification, costing in total $300 and working 7 days (8h). Additionally, API calls are necessary for analyzing the trajectories of existing GUI agents, which cost $100 in total. Overall, the benchmark construction costs $400, as shown in Table 13.

For data synthesis, we provide the cost analysis for rollouts on $20k$ tasks with parallel sampling 4, expansion rounds 32, which results in total 68 trajectories per task. For open-weights policy models, we deploy them on 32 NVIDIA A100 GPUs,

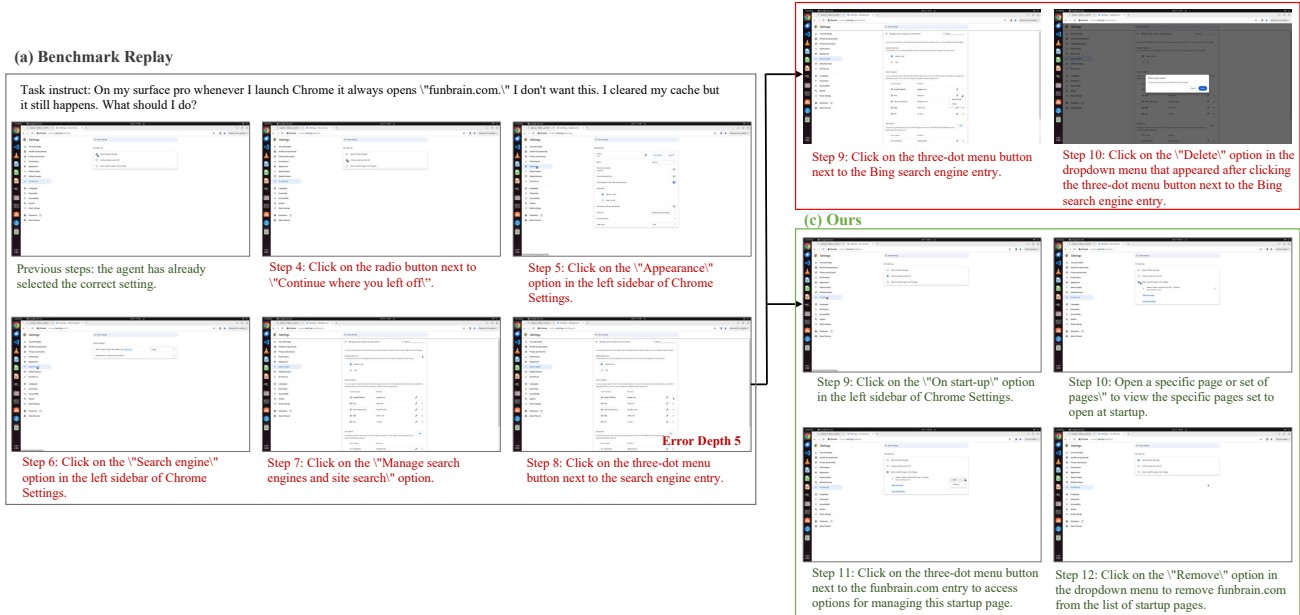

*Figure 17.* Trajectory Comparison of OpenCUA and RoTS Model on GUI-RobustEval.

Task Instruction: Find the FAQ page about ticket delivery.

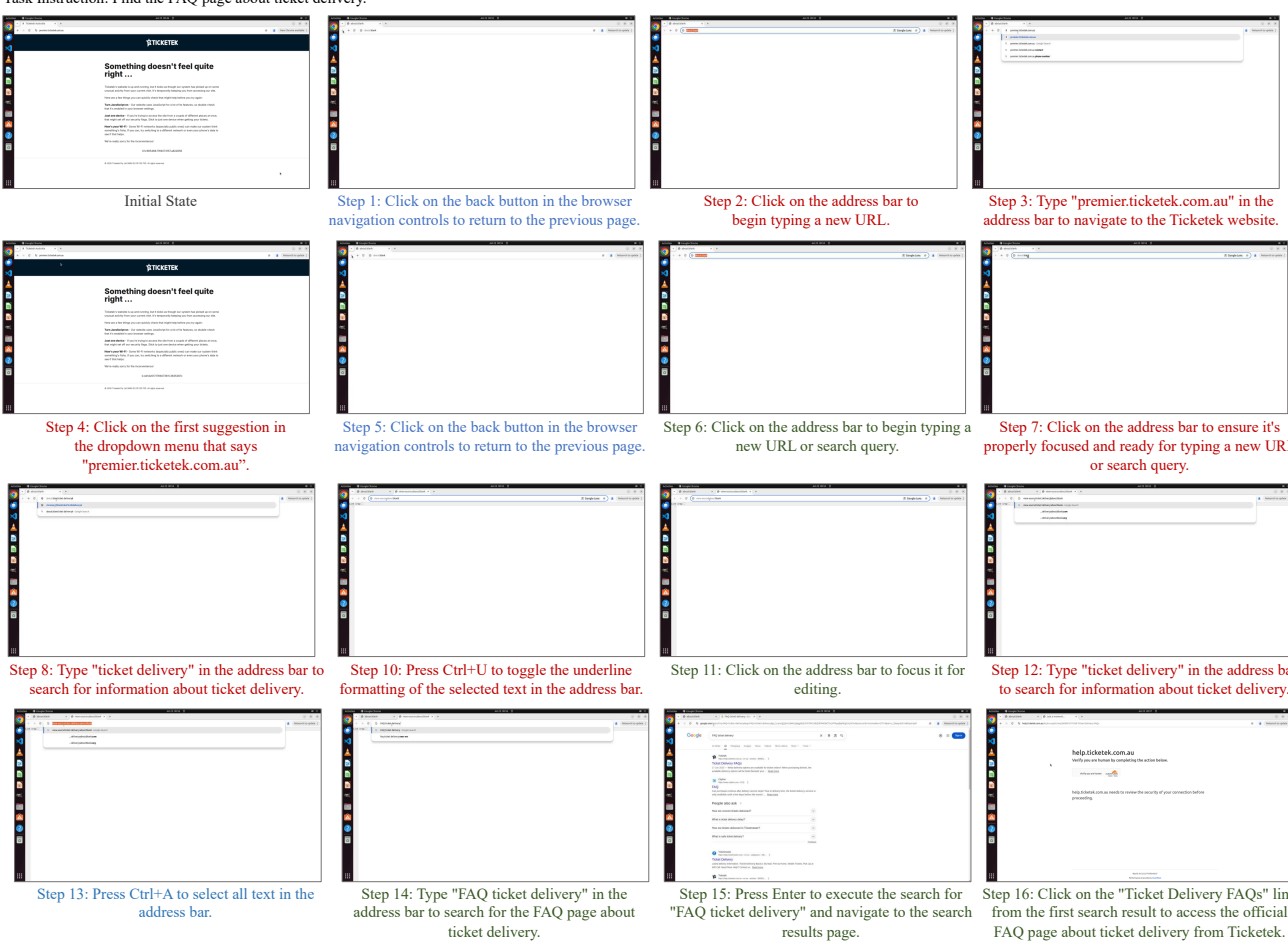

*Figure 18.* Exploration and Error Recovery Behavior of RoTS Model on OSWorld. Green text denotes correct actions, red text indicates failed attempts or erroneous actions, and blue text highlights steps involving error recovery.

while for proprietary models, we use APIs from corresponding provider. Thus, the cost comes from: the GPU server for self-deployed models, API calls and the Cloud Server for the environment deployment. As shown in Tables 14, the overall monetary cost is about **$48,100**, and the end-to-end wall-clock time is about **16 days** under the current configuration.

*Table 13.* Benchmark construction cost.

| Item | Cost (USD) | Time |
|---|---|---|
| Human filtering | 300 | 7 days |
| API calls | 100 | 2.05 hours |
| Total | 400 | 7 days |

*Table 14.* Sampled data generation cost.

| Item | Cost (USD) | Time |
|---|---|---|
| Self-deploy | 19,900 | 16 days |
| API calls | 21,700 | 16 days |
| Environment | 6,500 | 16 days |
| Total | 48,100 | 16 days |

## F.7. A Study on Experience-Informed Recovery

We conduct modular ablation on the effectiveness of experience-informed reflector $\pi_\theta^{er}$ and an advice-conditioned recovery actor $\pi_\theta^{rec}$ on GUI-RobustEval. We use Qwen3-VL-Plus as the original policy model and $\pi_\theta^{rec}$ and Qwen3-Max $\pi_\theta^{er}$.

As a baseline, we directly perform 20 parallel rollouts with the original policy to continue from the erroneous state, without reflection or advice. This consumes a similar rollout budget as EIR with 4 parallel rollouts and 32 rounds expansions. To construct the experience, then for each EIR variants, we first perform 4 parallel rollouts with original policy, and then

Task Instruction: Show me all men's large-size short-sleeve shirts with a discount of 50% or more.

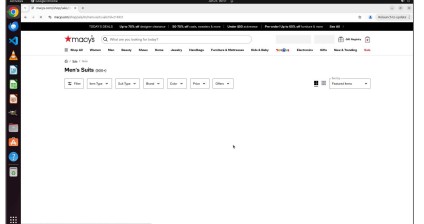

Agent response at this state: "Step 15:
## Thought:\nAfter clicking the \"Not Right Now\" button in the previous step, I expected to return to the main page, but instead I'm still on the \"Men's Suits on Sale\" page. This suggests that either the popup didn't close properly or we need to take additional steps to navigate back to our intended search results.\n\nLooking at our progress toward the goal of finding men's large-size short-sleeve shirts with 50% or more discount:\n1. We successfully searched for \"men large-size\"\n2. We clicked on the \"men's large size short sleeve shirts\" suggestion\n3. We navigated to the Men category\n4. We attempted to access sorting options\n5. We tried to go back to the search results but ended up on a different page\n\nTo get back on track with our goal, we need to navigate back to the search results page where we can refine our search to specifically look for large-size short-sleeve shirts with 50% or more discount. The back button in the browser would be the most direct way to return to the previous page.\n\nI notice that the back button is located in the upper left corner of the browser window. Clicking this button would take us back to the previous page, which should be the search results for \"men large-size\" or possibly the search results for \"men's large size short sleeve shirts\" that we were looking at before.\n\nAfter clicking the back button, I expect to return to the search results page where we can then refine our search further to specifically look for large-size short-sleeve shirts with 50% or more discount. This will help us get closer to our goal of finding the specific items we're looking for.
## Action:\nClick the back button in the upper left corner of the browser window to return to the previous page.
## Code:\n```python\npyautogui.click(x=90, y=88)\n``` "

Agent response at this state: " Step 16:
## Thought:\nAfter clicking the back button in the previous step, I expected to return to the search results page for men's large-size clothing items, but instead I'm still on the \"Men's Suits on Sale\" page. This suggests that either the back button didn't work as expected or we need to take additional steps to navigate back to our intended search results.\n\nLooking at our progress toward the goal of finding men's large-size short-sleeve shirts with 50% or more discount:\n1. We successfully searched for \"men large-size\"\n2. We clicked on the \"men's large size short sleeve shirts\" suggestion\n3. We navigated to the Men category\n4. We attempted to access sorting options\n5. We tried to go back to the search results but ended up on a different page\n\nTo get back on track with our goal, we need to navigate back to the search results page where we can specifically look for large-size short-sleeve shirts with 50% or more discount. The back button in the browser would be the most direct way to return to the previous page.\n\nI notice that the back button is located in the upper left corner of the browser window. Clicking this button would take us back to the previous page, which should be the search results for \"men large-size\" or possibly the search results for \"men's large size short sleeve shirts\" that we were looking at before.\n\nAfter clicking the back button, I expect to return to the search results page where we can then refine our search further to specifically look for large-size short-sleeve shirts with 50% or more discount. This will help us get closer to our goal of finding the specific items we're looking for.
## Action:\nClick the back button in the upper left corner of the browser window to return to the previous page.
## Code:\n```python\npyautogui.click(x=90, y=88)\n``` "

Agent response at this state: " Step 17:
## Thought:\nAfter clicking the back button in the previous step, I've successfully returned to the search results page for \"men's large size short sleeve shirts\" as expected. This is exactly what we needed since our goal is to find men's large-size short-sleeve shirts with a discount of 50% or more.\n\nLooking at the current state, I can see that we have several filtering options available to help us narrow down our search. Since our goal specifically mentions finding items with a 50% or more discount, the \"Price\" filter would be the most relevant option to use. By clicking on the Price filter, we'll be able to set a minimum price threshold that corresponds to a 50% discount (which would be $50 off for a $100 item).\n\nThe Price filter is currently showing a dropdown arrow, indicating that clicking on it will reveal various price range options. This is exactly what we need to proceed with our task. Once we click on the Price filter, we'll be able to select an appropriate price range that corresponds to a 50% or more discount.\n\nAfter setting the price filter, we'll likely need to apply other filters like sleeve length (which appears to already be set to \"Short Sleeve\") and potentially size to ensure we're only seeing large-size options. But first, let's focus on the price filter since it's the most critical aspect of our goal.\n\nI anticipate that clicking on the Price filter will open a dropdown menu with various price range options, allowing us to select a minimum price that corresponds to a 50% discount.
## Action:\nClick on the \"Price\" filter dropdown to access price range options for filtering the search results.\n\n
Code:\n```python\npyautogui.click(x=525, y=400)\n``` "

*Figure 19.* Over-reflection Behavior of the RoTS Model on OSWorld.

perform EIR for 32 times under the following setup: (i) Reflector (w/o exp.), reflector without trajectory-derived experience; (ii) Reflector (w/ exp.), reflector with trajectory-derived experience; (iii) EIR w/o advice, full recovery but removing advice conditioning; and (iv) Full EIR, the complete method. We report the averaged accuracy of error awareness rate and post-error success rate for all generated trajectories under each setting.

Table 15 shows that enabling the reflector improves error awareness (from 60.4 to 62.6), and providing experience further strengthens it (67.1); meanwhile, successful post-error recovery mainly comes from coupling the reflector with the advice-conditioned actor, where Full EIR achieves the best averaged post-error success rate 46.1. This module ensures that we can obtain effective long-horizon error recovery data during EIR round in tree expansion.

*Table 15.* Ablation of EIR on GUI-RobustEval. Exp.: experience-informed; $g$: advice. The results are averaged over total 32 trajectories.

| Variant | Recovery | Error Awareness Rate (%) | Post Error Success Rate (%) |
|---|---|---|---|
| No EIR (Base) | Actor (no $g$) | 60.4 | 42.9 |
| Reflector (w/o exp.) | – | 62.6 | n/a |
| Reflector (w/ exp.) | – | 67.1 | n/a |
| EIR w/o advice | Actor (no $g$) | 67.1 | 44.3 |
| Full EIR | Actor (+$g$) | 67.1 | 46.1 |

### F.8. Sensitivity to Recovery Depth in Training Dataset

To isolate the effect of recovery horizon on model perfor-mance, we conduct a depth-cap ablation under the $100k$/7B setting. We filter $\mathcal{D}_{\text{ref}}$ by capping the maximum recov-ery depth (*i.e.*, the number of steps from the error state to task completion) while keeping the total data budget and $\lambda_{\text{ref}} = 0.1$ fixed. As shown in Table 16, the largest marginal gain comes from moderate-depth recovery data (depth $\leq 5$), which improves post-error success from 12.1 to 20.7 (+8.6). Gains beyond depth 7 are relatively small (+0.8), suggesting that while long-horizon recovery supervi-sion is beneficial, the most cost-effective region is moderate depth. This finding is consistent with the expansion-round analysis in Fig. 5(a), where deeper co-expansion naturally harvests longer recovery traces.

*Table 16.* Ablation on maximum recovery depth in $\mathcal{D}_{\text{ref}}$. We filter reflection-related data by capping the recovery depth while keeping total budget and $\lambda_{\text{ref}} = 0.1$ fixed.

| Max Depth | GUI-RobustEval(%) | | OSWorld (%) | |
|---|---|---|---|---|
| | Post. Succ. | AP@4 | AP@4 | Succ. |
| No $\mathcal{D}_{\text{ref}}$ | 12.1 | 8.0 | 8.6 | 18.5 |
| $\leq 2$ | 15.5 | 10.5 | 10.0 | 19.5 |
| $\leq 5$ | 20.7 | 13.4 | 12.8 | 21.1 |
| $\leq 7$ | 21.5 | 13.8 | 13.5 | 21.3 |
| **No cap** | **22.1** | **14.1** | **14.1** | **21.4** |

