# OpenReview forum: "Recovering Policy-Induced Errors: Benchmarking and Trajectory Synthesis for Robust GUI Agents"
_ICML.cc/2026/Conference — ICML 2026 spotlight_

### Official Review · Reviewer_cZ3L · 2026-03-12

**Soundness:** 3
**Presentation:** 3
**Significance:** 3
**Originality:** 3
**Overall Recommendation:** 4
**Confidence:** 4

**Summary:**

This paper proposes GUI-RobustEval, a benchmark for evaluating recovery from policy-induced errors in GUI agents, and RoTS, a tree-based trajectory synthesis pipeline for generating long-horizon failure-recovery data. The paper argues that existing benchmarks and training data do not match the real distribution of policy-induced errors, especially in terms of error types and error depth. To address this, the authors construct 1,216 test cases across 12 error types and 4 error depths, and synthesize 800k training samples with an explore-recovery co-expansion pipeline.

**Compliance With Llm Reviewing Policy:**

Affirmed.

**Final Justification:**

The paper shows a reasonable level of novelty, and I keep my original score.

**Key Questions For Authors:**

How is WinDeskGround fundamentally different from a well-designed synthetic stress test, beyond being the first one focused on multi-window desktops?

Why is 85% instruction-validity approval considered sufficient for a benchmark of this type? That still means a noticeable fraction of instructions may be unclear or imperfect.


The semantic interference curves in Figure 4(c) are quite flat. Does this mean the similarity-based distractor design is not actually very challenging?


Why are the distractor windows often placed behind the target if the goal is to test semantic confusion more directly?

**Limitations:**

The paper studies an important problem, but the current benchmark still feels somewhat narrow. The scenes are synthetic and static, the dataset is not especially large, and some of the key interference factors look weaker than advertised. The paper is useful as a stress test, but not yet fully convincing as a realistic benchmark of desktop robustness.

**Strengths And Weaknesses:**

## Strengths
1、The paper studies a meaningful problem. Robust recovery from the agent’s own mistakes is clearly important for real GUI deployment.

2、The benchmark contribution is useful. Measuring error awareness and post-error success is more targeted than reporting only final task success.

3、The paper does more than introduce a benchmark. It also proposes a concrete synthesis pipeline and scales it to 800k training samples.

## Weaknesses

### Weaknesses:

“Policy-induced error recovery” is a good angle, but the main method still looks quite close to benchmark curation and reflection-style data synthesis, what is really new here beyond combining exploration, replay, and recovery in one pipeline?

The evaluation is narrow: everything is built around desktop GUI tasks, and GUI-RobustEval has only 1,216 cases, this seems limited for the scope of the claims.

The gain on GUI-RobustEval is real but not that large, if RoTS-32B reaches 40.3 average post-error success while strong models like Qwen3-VL-Plus are already at 33.5 at depth 5, is the improvement big enough for such a heavy synthesis pipeline?

The paper fixes N = 4, 32 expansion rounds, λ_ref = 0.1, and the exact UCB-style explore/recover rules, but there is little sensitivity analysis, these seem like central choices, not minor details.

---

> ### Author Rebuttal · Authors · 2026-03-31
>
> We thank the reviewer for the positive assessment and address each concern below.
>
> ---
>
> ## W1: what is really new beyond combining exploration, replay, and recovery?
> We appreciate this question. The novelty of RoTS lies in a **problem-driven design** addressing two empirical gaps: **error-coverage mismatch** and **error-horizon mismatch** from evaluation and data perspective.
> 1. GUI-RobustEval goes beyond standard benchmark curation by introducing a new error source (policy-induced, not synthetic/adversarial), a new evaluation dimension (depth-controlled recovery), and fine-grained diagnostics (awareness + success × type × depth), providing rich signals for diagnostics.
>
> 2. Methodologically, RoTS introduces two specific innovations over standard "run-then-reflect" pipelines:
> - Rather than passively correcting errors from existing trajectories, FDE proactively discovers failure modes that parallel sampling misses. Table 4: FDE contributes +1.5 OSWorld success and +2.3 GUI-RobustEval success over parallel sampling.
>
> - EIR uses neighbor-branch experiences to generate advice-conditioned recovery rollouts to improve effective error recovery trajectories. Table 12: reflector with experience outperforms without (67.1% vs. 62.6% awareness); advice conditioning improves recovery (46.1% vs. 44.3%).
>
> In summary, the novelty lies in identifying two empirical gaps and a synergistic design (GUI-RobustEval + RoTS) to close them with measurable gains.
>
> ---
>
> ## W2: Evaluation scope — 1,216 desktop cases
> We agree broader coverage strengthens benchmarks, and address both aspects below.
>
> **On 1,216 cases:** GUI-RobustEval is an online executable benchmark requiring human annotation, replay checks, and controlled-depth evaluation. For context, comparable online benchmarks are smaller: OSWorld (369), D-GARA (152). Importantly, the benchmark is grounded in failures from 12 SOTA agents and 1.5k trajectories, covering 12 error types × 4 depths for fine-grained diagnosis.
>
> **On desktop only:** Desktop is a challenging setting; leading benchmarks (OSWorld, WindowsAgentArena) similarly focus on it. Our framework is architecture-agnostic and it does not depend on desktop-specific features. Extending to mobile requires only new task curation, not method changes, and we note this in Section 6 (limitation).
>
> We will clarify the benchmark's positioning and scope in the revised paper.
>
> ---
>
> ## W3: Performance Gains
> We appreciate this question. The key advantage of RoTS lies in **robustness beyond single-depth accuracy**. And we present evidence from three perspectives below.
>
> **(a) Robustness gains are substantial:** As shown in Tables 2 & 3, RoTS-32B improves All-Pass@4 by +9.3 points over Qwen3-VL-Plus (proprietary, 38% relative), with a smaller depth degradation (↓33% vs. ↓39%), showing the effectiveness of RoTS to improve error-recovery ability of agents.
>
> **(b) Design-driven, not compute-driven.** Under a matched rollout budget (Table 4), FDE+EIR improves post-error success (12.1→22.1) and All-Pass@4 (8.6→14.1) over PS.
>
> **(c) Gains generalize across backbones:** We conduct a controlled experiment applying RoTS to the Qwen3-VL-8B backbone, which yields +11.4 pt All-Pass@4 (19.6→31.0, 58% relative), reduces depth-5 degradation from 48% to 34%, and obtaining stronger performance on a better backbone.
>
>
> ---
>
> ## W4: Sensitivity analysis
>
> We appreciate this point. The main paper provides sensitivity analysis for several key hyperparameters: **λ_ref** (Fig. 4) sweeps 6 values with an optimum at 0.1; **expansion rounds** (Fig. 5(a)) show monotonic improvement from 0 to 32; **dataset size** (Fig. 5(b)) shows diminishing returns beyond 800k.
>
> For the remaining choices, we provide additional experiments below.
>
> **N (initial parallel rollouts):** We vary N under fixed co-expansion and find performance stable across 2–8, with N=4 balancing coverage and cost:
>
> |N|GRE Post. Succ.|OSWorld Succ.|
> |---|---|---|
> |2|20.8|20.6|
> |4|22.1|21.4|
> |8|22.6|21.8|
>
> Under this setting, the small variance (~5% relative) indicates diminishing returns beyond N=4, supporting it as a practical default.
>
> **UCB exploration constant c:** We train models on data synthesized with different node selection strategies under a 7B/100k. Results on GUI-RobustEval and OSWorld:
>
> |Selection Strategy|GRE Post. Succ.|OSWorld Succ.|
> |---|---|---|
> |Greedy (c=0)|18.5|18.8|
> |UCB (c=0.25)|22.1|21.4|
> |UCB (c=0.5)|21.6|21.1|
> |Random|17.2|18.1|
>
> UCB-based selection outperforms both Greedy and Random, confirming that balanced exploration-exploitation is essential for synthesizing high-quality training data.
>
> ---
>
> ## Key Questions
>
> Regarding the listed key questions, we may not have fully understood the intended references. Some of the mentioned terms and figure references do not directly correspond to our submission. If the reviewer intended related concerns about submission, we would be very happy to clarify them in the revision and address the corresponding points more explicitly.

---

> > ### Author Rebuttal · Reviewer_cZ3L · 2026-04-06
> >
> > Thanks for answering the question. I will keep the score.

---

### Official Review · Reviewer_ce5r · 2026-03-12

**Soundness:** 4
**Presentation:** 3
**Significance:** 4
**Originality:** 4
**Overall Recommendation:** 5
**Confidence:** 4

**Summary:**

This paper focuses on improving agent's robustness to policy-induced errors by introducing a new benchmark, GUI-RobustEval that contains 1,216 test cases of 12 error types and 4 error depths to evaluate agents' ability to be aware of an error they've encountered and their ability to recover from errors. Additionally, this paper introduces an 800k dataset through a data synthesis pipeline, Robustness-driven Trajectory Synthesis (RoTS), a tree-search based algorithm that prunes trajectories into a successful subtree to explore in high risk states where the policy is likely to choose the wrong action (fragility-driven exploration) and a failed subtree to explore the most promising recovery actions to correct from an error (experience-informed recovery). Altogether, finetuning a Qwen2.5-VL-7B and Qwen2.5-VL-32B leads to strong error-awareness and error-recovery performance on GUI-RobustEval as well as competitive performance on OSWorld in terms of success rate and All-Pass@4.

**Compliance With Llm Reviewing Policy:**

Affirmed.

**Key Questions For Authors:**

Overall this paper does a great job with strong empirical evidence for the effectiveness and quality of their dataset and also puts forth a benchmark for policy-induced errors that has been extensively annotated. My questions will focus on the failure analysis in GUI-Robust.

(1) How does the error awareness correlate with the success rate on GUI-RobustEval? For example, in Table 2 GPT 5.1 and Jedi-7B w/ GPT 5.1 appear to be about equally capable at detecting errors but not necessarily recovering from them.

(2) What does the per-error type error awareness and error success rate look like for RoTS? It would be interesting to analyze how the RoTS dataset contributes to addressing each error type and where the bottleneck lies (likely progress perception and planning) in the data synthesis pipeline for future work directions.

(3) In both GUI-Robust and RoTS an implicit assumption is made that trajectories and nodes are replayable given the root-to-node prefix; however, this isn't necessarily true. For parametrized tasks (such as "Book this flight N days for now") or stochastic environments (e.g. pop-up ads or bot detectors on web) a trajectory is not guaranteed to be replayable. Can you clarify what is meant by "replayable" and how these tasks would be handled?

**Limitations:**

yes

**Strengths And Weaknesses:**

$Soundness$: The paper presents strong experimental results and ablations that justify RoTS' design decisions as well as including equal rollout budget experiments and data quality analyses.

$Presentation$: The paper differentiates GUI-RobustEval clearly from past benchmarks that investigate GUI robustness. The figures are pretty dense (particularly Figure 1 and Figure 2) and can be clarified. It is unclear what the bottom part of Fig (a) and (b) are representing (is this what RoTS synthesizes?). Figure 2 devotes more space to GUI-RobustEval which is fairly straightforward to follow but compresses RoTS into a tight diagram that includes keywords but otherwise is very difficult to understand even after reading the paper.

$Significance$: This paper tackles errors induced by the agent itself which are very require careful annotation and are difficult to collect data for, all which is handled through the introduction of a new benchmark and data synthesis pipeline.

$Originality:$ There are plenty of works on GUI robustness but this paper focuses on policy-induced errors which is novel compared to previous works and additionally includes a sizable dataset for open-source training.

---

> ### Author Rebuttal · Authors · 2026-03-31
>
> We thank the reviewer for the positive and detailed assessment. We address each question below.
>
>
> ## Q1: How does error awareness correlate with success rate?
>
> Thank you for this question. In GUI-RobustEval, error awareness measures whether the agent, immediately at takeover, recognizes that the injected prefix has led to an erroneous state; it does not measure whether the agent can subsequently recover. Therefore, awareness and post-error success are positively correlated but not equivalent: awareness is a prerequisite, while recovery additionally requires re-planning, grounding, and multi-step execution.
>
> This explains the GPT 5.1 vs. Jedi-7B w/ GPT 5.1 case in Table 2. At takeover, both receive the same erroneous history and Jedi's planner is also GPT 5.1, so awareness rates are similar (~34%). However, post-error success diverges (16.1% vs 40.3%): GPT 5.1 must jointly reason and act end-to-end, whereas Jedi separates planning and grounding, converting similar awareness into more stable recovery actions. RoTS improves both channels: compared to OpenCUA-7B (awareness 46.3%, success 28.3%), RoTS-7B raises awareness to 51.9% and success to 34.2%. This is exactly why GUI-RobustEval reports both metrics — one tests error perception, the other tests action capability.
>
> ---
>
> ## Q2: Per-error-type analysis for RoTS
>
> Thank you for this suggestion. We agree that a per-error-type breakdown is very informative. We provide the per-error-type breakdown for RoTS-32B on GUI-RobustEval (averaged across depths). Due to space, we report post-error success here; we have also computed the corresponding per-error-type awareness breakdown and will include both metrics in the revision.
>
> |Error Type|Baseline Avg. Succ.*|RoTS-32B Succ.|Improvement|
> |---|---|---|---|
> |Incorrect UI Element|43.1%|52.3%|+9.2|
> |Grounding Failure|35.2%|42.7%|+7.5|
> |Ineffective Action|42.6%|53.9%|+11.3|
> |Typing Error|45.4%|54.8%|+9.4|
> |Miss Necessary Step|28.6%|38.2%|+9.6|
> |Incorrect Tool Usage|23.1%|30.8%|+7.7|
> |Wrong Target|12.7%|20.9%|+8.2|
> |Incorrect Parameter|22.8%|31.4%|+8.6|
> |Misunderstand Task Objective|16.7%|24.1%|+7.4|
> |Fail to Terminate|11.1%|17.4%|+6.3|
> |Lack of Knowledge|22.9%|33.4%|+10.5|
>
> *(\*Baseline average from Table 6, averaged over 5 SOTA agents.)*
>
> **Key findings:**
> - RoTS brings broad improvements across most error types rather than only a narrow subset.
> - Bottlenecks remain in Fail to Terminate (17.4%) and Misunderstand Task Objective (24.1%), where recovery requires accurate progress perception and re-planning over long histories — important directions for future work.
>
> We will add this analysis in the revision.
>
> ---
>
> ## Q3: Replayability assumption
> Thank you for raising this important point. We agree that, in real open-world settings, a node is not always replayable from the root-to-node prefix, especially for tasks with dynamic parameters (e.g., “book this flight N days from now”) or external stochasticity.
>
> In our paper, "replayable" means that given the same system snapshot, task initialization config, and root-to-node action prefix, the environment can be restored to a semantically equivalent state for continued rollout/evaluation — not strict equality in all open-world cases.
>
> **For GUI-RobustEval (stricter guarantee):** We use unified base images with fixed initialization scripts, disable reproducibility-breaking factors (pop-ups, updates, window-state changes), and perform manual replay verification — only keeping examples whose error states are stably reproducible in our setup. For web tasks, we prioritize websites with stable interaction flows.
>
> **For RoTS training data (approximate):** We can not gurantee that 20k training tasks are strictly replayable, therefore we adopt an "environment control + posterior filtering" strategy. Beyond the same environment controls as GUI-RobustEval, after replay and tree expansion, as described in Sec. 3.5, we use VLM-based critics to check whether actions, state transitions, and screenshots are consistent, and filter out mismatched or abnormal trajectories.
>
> Replayability is an applicability condition and limitation of our benchmark/synthesis pipeline, not a property assumed for arbitrary GUI tasks. We will clarify this in the revision.
>
> ---
>
> ## Q4: Clarifying Figure 1 and Figure 2
>
> Thank you for this suggestion and we agree the figures are too dense. For the bottom parts of Fig. 1(a)(b): they represent reflection-related data synthesized by RoTS. The upper parts show agents failing to recover from diverse delayed errors; the bottom parts show RoTS synthesizes recovery supervision matching this distribution. In the revision, we will: (1) add explicit labels separating "observed failures" from "RoTS-synthesized data"; (2) revise the caption to explain this contrast directly; (3) redesign Figure 2 to give the RoTS pipeline more space with clearer stage decomposition (replayable environment, FDE, EIR, dataset construction).

---

> > ### Author Rebuttal · Reviewer_ce5r · 2026-04-04
> >
> > Thank you for the response. The rebuttal has addressed my remaining scores and I will be maintaining my strong score.

---

### Official Review · Reviewer_d9gV · 2026-03-13

**Soundness:** 3
**Presentation:** 3
**Significance:** 3
**Originality:** 3
**Overall Recommendation:** 5
**Confidence:** 4

**Summary:**

This paper studies robustness of GUI agents to policy-induced errors, meaning errors caused by the agent’s own actions during execution rather than exogenous perturbations. The authors contribute GUI-RobustEval, an online benchmark with 1,216 executable test cases spanning 12 error types and 4 error depths, and RoTS, a tree-based trajectory synthesis pipeline that explores fragile states and generates recovery trajectories. Using 800k synthesized samples, they fine-tune 7B and 32B Qwen2.5-VL models and report improvements on GUI-RobustEval as well as OSWorld and WindowsAgentArena, with RoTS-32B reaching 47.4% success and 33.8% All-Pass@4 on OSWorld.

**Compliance With Llm Reviewing Policy:**

Affirmed.

**Final Justification:**

see ack

**Key Questions For Authors:**

1)Are the 20k synthesis tasks fully disjoint from GUI-RobustEval, OSWorld-Verified, and WindowsAgentArena? Please answer at the level of tasks, environment snapshots, websites/files/documents, and any reused assets. A clear yes, with specifics, would significantly increase my confidence.

2)How is Error-Awareness Rate operationalized across heterogeneous agents, and why are some awareness entries in Table 2 missing? If you can show that the awareness metric is robust to output-style differences, my opinion would improve.

**Limitations:**

yes

**Strengths And Weaknesses:**

Strengths.

1)The paper targets a real weakness of current GUI agents, namely their inability to notice and recover from their own mistakes, rather than only handling external noise or adversarial perturbations.

2)GUI-RobustEval is a meaningful benchmark contribution. The idea of evaluating from erroneous prefixes with controllable depth is well-motivated and more diagnostic than raw task success.

3)The benchmark and the synthesis pipeline are aligned. This is not just “here is a benchmark” plus “here is some training data”; the paper makes a coherent argument that the data generation procedure is designed to match the failure modes the benchmark measures.

Weaknesses.

1)The system is relatively complex, relying on multiple judge/critic modules. The coexistence of components such as the reward model, progress critic, action critic, reflection identifier, and reflector makes it difficult to determine precisely which part contributes to the final improvement and how sensitive the system is to errors in individual modules, necessitating further in-depth analysis.

2)The paper never clearly states that synthesis data is disjoint from evaluation tasks and assets. The main results in Table 2 and Table 3 are much easier to trust if the paper explicitly states that GUI-RobustEval, OSWorld-Verified, and WindowsAgentArena are disjoint from the synthesis pool at the task, snapshot, and asset levels. Without that, an uncomfortable question hangs over the whole empirical section.

3)The fairness story across heterogeneous agents is not fully convincing. On Page 3, the benchmark normalizes each step to a standard executable form and then converts it back to each agent’s native format. That is practical, but it is not neutral by default. Different agents may rely on different history formats, internal reasoning conventions, or action abstractions, so the translation layer itself can advantage or disadvantage them. The paper should say more about how this conversion was validated. Otherwise the benchmark could be partly measuring interface compatibility.

4)The ablations do not fully bridge to the main result setting. The ablations in Table 4 and Table 5 are on 100k data and a 7B model, while the main headline results use 800k data and include a 32B model. The gap is large enough that the reader is still asked to extrapolate. A strong paper would include at least one reduced but full-scale control closer to the headline setting.

5)The strongest causal interpretation is overstated. The paper repeatedly implies that improved long-horizon recovery ability is what drives better conventional benchmark performance. The evidence is consistent with that view, but it does not prove it. Table 5 and Table 3 show that RoTS-trained models do better, not that recovery ability is the sole or dominant causal channel.

---

> ### Author Rebuttal · Authors · 2026-03-31
>
> We thank the reviewer for the positive assessment and address each concern below.
>
> ---
>
> ## Q1 & W2: Are the 20k synthesis tasks fully disjoint from evaluation?
>
> **Yes.** The 20K RoTS synthesis tasks and all evaluation benchmarks are disjoint at task and asset level. We share base OS snapshots (common in GUI-agent research) but not benchmark-specific tasks or assets. Our synthesis pool comprises 10K AgentNet tasks with reconstructed instances and 10K self-curated persona-driven tasks (Appendix B.2). We will clarify train/eval separation in the revision.
>
> ---
>
> ## Q2: Error-Awareness Rate across heterogeneous agents
>
> Thanks for the insightful question. In common GUI agent, the agent's output contains 3 parts: thought, action summary, and executable code. Error-Awareness is judged from the first thought after agents' takeover, as it is where agents express recognition of prior errors. We use Qwen3-VL-Plus as judge model and evaluate human agreement on 200 samples from GUI-RobustEval. We summarize the result from 3 agents that have different thought style:
>
> |Model|Agreement (%)|F1|
> |-|-|-|
> |Qwen3-VL-Plus|97.0|96.2|
> |UI-TARS-1.5|96.0|95.0|
> |OpenCUA-7B|97.0|96.0|
>
> The judge achieves comparable agreement and F1 scores on different GUI agents, confirming the robustness to output-style differences.
>
> **Missing entries.** The "−" entries (Qwen2.5-VL-7B, Qwen3-VL-8B) output only action and code without thought during evaluation. Since our awareness metric relies on thought to detect error recognition of agents, judging from action summary might be unreliable. We omit these rather than report misleading numbers, and acknowledge this as a limitation of error awareness.
>
> ---
>
> ## W1: System complexity — which module contributes?
> Thank you for raising this point. We clarify the role of each module in the pipeline: tree partitioning (reward model), FDE node selection (progress critic), EIR error localization and recovery (reflector + recovery actor), and data filtering (action critic + progress critic). The first three are core generation; the last is quality control.
>
> **(a) Data generation pipeline ablation**: Because of this grouping, the data generation pipeline ablation in Table 4 implicitly tests each core module's contribution. On GUI-RobustEval, FDE (progress critic) adds +2.3% over PS baseline (12.1→14.4), EIR (reflector + recovery actor) adds +6.0% (12.1→18.1), and combining both yields +10.0% (12.1→22.1).
>
> **(b) EIR ablation** (Table 12): Zooming into EIR, Reflector improves Error Awareness (+6.7%, 60.4→67.1); adding actor further improves Post-Error Success (+3.2%, 42.9→46.1).
>
> **(c) Module reliability:**
> We evaluate human agreement on sampled data (200 instances each) to study the reliability of each critic module:
>
> |Module|Agreement (%)|F1|
> |-|-|-|
> |Action critic|90.6|90.4|
> |Progress critic|88.7|88.7|
>
> Both critics show ≥88% agreement with human judgment. As downstream filters, they limit noise from upstream modules before it enters the training set. We will clarify module roles and add these validations in the revision.
>
> ---
> ## W2: Train/eval separation
> Please refer to Q1 & W2.
>
> ---
>
> ## W3: Fairness of benchmark format conversion
>
> This is a valuable concern. We agree the format conversion is not perfectly neutral and may advantage or disadvantage certain agents, as the reviewer noted. This is a trade-off of our benchmark design: to evaluate error recovery, we need to place agents into a specific erroneous state, which requires injecting a prefix history — and that inevitably involves cross-format conversion. That said, since the same converted format is applied consistently across all depths for a given agent, we expect format artifacts to affect all depth levels similarly rather than distort the within-agent degradation trend. We will discuss this limitation explicitly in the revision.
>
> ---
>
> ## W4: Ablation scale gap
>
> We use 100K/7B to isolate *which design choices matter* under controlled budget, then evaluate at full scale. Fig. 5(b) shows the RoTS gain persists from 50K to 1000K under 7B.
>
> To bridge more directly, we add two closer-to-headline controls:
>
> |Setting|OSWorld (%)|RobustEval Avg. (%)|
> |-|-|-|
> |7B, 800k: PS|31.5|20.5|
> |7B, 800k: PS+FDE+EIR|36.3|34.2|
> |32B, 100k: PS|22.1|15.6|
> |32B, 100k: PS+FDE+EIR|27.8|26.0|
>
> These verify that ablation conclusions generalize to larger data scale and larger backbone.
>
> ---
>
> ## W5: Overstated causal interpretation
>
> We agree our experiments do not prove recovery is the sole causal channel. Our intended claim is narrower: RoTS primarily improves robustness to policy-induced errors, reflected in GUI-RobustEval and All-Pass@4. While RoTS also achieves improvement on standard benchmarks like OSWorld, we recognize that improved robustness is likely one of several contributing factors — better task, richer exploration and better CoT quality may also play a role. We will revise the paper to avoid overly strong causal wording and clarify this scope.

---

> > ### Author Rebuttal · Reviewer_d9gV · 2026-04-02
> >
> > The authors have addressed the concerns regarding the disjoint nature of the synthesis tasks and evaluation benchmarks, clarifying that they are indeed disjoint at the task, environment snapshots, and asset levels, which significantly strengthens the empirical claims. Furthermore, the explanation of how error-awareness is operationalized and the missing entries in the table were clarified. The authors also confirmed the robustness of the error-awareness metric across different agent output styles, which resolves the issue. Therefore, I believe these clarifications warrant an increase in the score.

---

### Official Review · Reviewer_g97p · 2026-03-23

**Soundness:** 3
**Presentation:** 3
**Significance:** 3
**Originality:** 2
**Overall Recommendation:** 4
**Confidence:** 3

**Summary:**

This work presents an effort on benchmark construction and a data synthesis pipeline for existing GUI agents to better diagnose policy-induced errors.  The authors first identify two key mismatches in the current evaluation framework, i.e. failure types and planning horizons. Both serve as strong motivation to build compositional and long-horizon evaluation for the better GUI agents benchmark that ties more directly with its real-world deployments. The authors first introduce GUI-RobustEval and ROTS, a comprehensive benchmark and an automatic data pipeline, which results in a large-scale, high-quality training corpus of approximately 800K samples. The authors then conduct comprehensive experiments on that models trained with RoTS-synthesized data. These models achieve substantial improvements in the overall task success rate, showing an empirical validation for the importance of modeling long-horizon reflection and recovery in GUI agents.

**Compliance With Llm Reviewing Policy:**

Affirmed.

**Final Justification:**

My concerns have been adequately addressed by the authors in the rebuttal. Will keep my positive recommendation.

**Key Questions For Authors:**

**Performance gains vs. method complexity**: In Table 2, RoTS performs comparably to Qwen3 at matched model scale without consistent improvement. Can the authors provide additional results, e.g., in more diverse task settings or in larger-scale experiments that demonstrate clear advantages over Qwen3? A convincing demonstration of meaningful performance margins would significantly strengthen my evaluation of the method's practical impact.

**Computational and data cost of RoTS**: RoTS relies on large-scale tree rollouts and multi-round co-expansion. Could the authors provide a detailed cost breakdown (e.g., average rollouts per task, data generation time, GPU/CPU consumption) and analyze how resource overhead scales with increasing error depth? Evidence of a favorable cost-performance trade-off, or a more efficient approximation, would substantially improve my assessment of the method's practicality.

**Ablation over error horizon**: The paper emphasizes long-horizon error recovery but lacks systematic ablation over error depth. Does model performance continue to improve beyond depth >5, and is there a trade-off between performance and generalization at greater depths? This analysis would clarify the method's applicable scope and improve the overall completeness of the paper.

**Baseline coverage**: The current comparisons focus on standard GUI agent models, with limited engagement with recent long-horizon or exploration-based methods such as WebExplorer [1], AndroTMem [4], and related works [2,3]. Have the authors evaluated these methods on GUI-RobustEval, or can they discuss how RoTS differs from these approaches in modeling long-horizon behavior? Including such comparisons or discussion would sharpen the paper's positioning within the current research landscape.


> [1] Liu, Junteng, et al. "Webexplorer: Explore and evolve for training long-horizon web agents." 2025.
>
> [2] Guo, Yuan, et al. "Atomic-to-Compositional Generalization for Mobile Agents with A New Benchmark and Scheduling System." 2025
>
> [3] Deng, Zehao, et al. "Training High-Level Schedulers with Execution-Feedback Reinforcement Learning for Long-Horizon GUI Automation." 2025
>
> [4] Shi, Yibo, et al. "AndroTMem: From Interaction Trajectories to Anchored Memory in Long-Horizon GUI Agents." 2026

**Limitations:**

Yes.

**Strengths And Weaknesses:**

**Strengths.**

- **Soundness**: Technically solid pipeline with clear problem formulation. The presented closed-loop system around policy-induced errors (GUI-RobustEval + RoTS) is well-motivated, and design choices (tree-based co-expansion, FDE, EIR) are adequately validated by experiments and ablation studies.
- **Presentation**: The paper is generally well-written and easy to follow. The authors have demonstrated benchmark construction and method details are described with sufficient specificity to support reproducibility.
- **Significance**: Targets a practically important and widely-acknowledged challenge in real-world GUI agent deployment. This work has potential to encourage more data-centric research for GUI agents.
- **Originality**: Core components build on established techniques (tree search, self-training, reflection), but their systematic integration and the introduction of the policy-induced error provide a meaningful and distinguishing conceptual contribution.


**Weaknesses.**

- **Limited performance gains**: The method performs comparably to Qwen3 in key comparisons (e.g., Table 2), and does not consistently outperform Qwen3 at matched model scale, weakening the persuasiveness of the claimed improvements.
- **Insufficient cost analysis**: The paper lacks adequate discussion of the computational and data costs of RoTS, particularly the trade-offs when increasing error depth (rollout overhead, data generation time, potential impact on generalization).
- **Incomplete ablation**: Despite emphasizing long-horizon error recovery, the paper does not systematically ablate the effect of varying error horizons on performance vs. cost, limiting understanding of the method's applicable scope.
- **Evaluating more relevant baselines in the benchmark**: some recent works [1-4] show similar observations in expanding the agents to long-horizon exploration. It would be interesting to see if the authors can evaluate some of these efforts in the current benchmark and discuss some more findings.

> [1] Liu, Junteng, et al. "Webexplorer: Explore and evolve for training long-horizon web agents." arXiv 2025.
>
> [2] Guo, Yuan, et al. "Atomic-to-Compositional Generalization for Mobile Agents with A New Benchmark and Scheduling System." arXiv 2025
>
> [3] Deng, Zehao, et al. "Training High-Level Schedulers with Execution-Feedback Reinforcement Learning for Long-Horizon GUI Automation." arXiv 2025
>
> [4] Shi, Yibo, et al. "AndroTMem: From Interaction Trajectories to Anchored Memory in Long-Horizon GUI Agents." arXiv 2026

---

> ### Author Rebuttal · Authors · 2026-03-30
>
> We thank the reviewer for the positive assessment and address each concern below.
>
> ## Q1: Performance gains vs Qwen3
> This is a valuable observation. We agree that Qwen3 is trained on substantially more in-house data, which gives it a higher depth-0 baseline (Qwen3-VL-8B: 48.1 vs. RoTS-7B: 43.5) on GUI-RobustEval. However, as error depth increases, Qwen3-VL-8B degrades far more sharply (↓48% vs ↓38%). This supports our core contribution: scaling general-purpose GUI data cannot match the distribution of policy-induced errors—precisely the gap RoTS is designed to fill.
>
> More importantly, RoTS is a data-centric method orthogonal to the base model. Thus we further train Qwen3-VL-8B with the same training recipe and summarize the results below:
>
>
> |Model|Depth-0|Depth-5|Drop (%)|OSWorld Succ. (50)|All-Pass@4|
> |-|-|-|-|-|-|
> |Qwen3-VL-8B|48.1|25.0|↓48|33.9|19.6|
> |RoTS-8B (Qwen3)|50.6|33.4|↓34|42.3|31.0|
>
> This controlled comparison shows that applying RoTS to the stronger Qwen3-VL-8B backbone achieves consistent gains across all error depths and robustness metrics.
>
> ## Q2: Computational and data cost of RoTS
> **Cost breakdown**
>
> We provide a detailed cost breakdown in Appendix F.5 (Tables 10–11). RoTS is run on 20K tasks with 3 policy models. For each task-policy pair, we use 4 initial rollouts and 32 co-expansion rounds, giving 68 trajectories, taking 16 days:
>
> |Component|Cost (USD)|
> |-|-|
> |Self-deployed GPU (Policy model)|19,900|
> |API calls (RM, Post-Proc, etc.)|21,700|
> |Cloud environment (Environment)|6,500|
>
>
> This pipeline produces 2.21M post-processed trainable samples (Table.8), \\$0.022/sample. To clarify, 2.21M is the full generated trainable corpus, while 800K is the controlled subset used in the main experiments. For comparison, OpenCUA's human-annotated dataset (22K tasks, 409K samples, 6 months, 634 annotators) costs ~\\$0.078/sample while excluding error recovery scenarios that are difficult to annotate manually.
>
> ---
>
> **Resource overhead vs error depth and Cost-performance trade-off**:
>
> This is an insightful point. In RoTS, it does not require separate targeted rollouts for each error depth: deeper recovery traces are harvested as expansion increases, so the dominant cost is controlled by expansion rounds rather than by targeted larger depths. In Fig. 5(a), performance improves monotonically with expansion rounds, with gains beginning to saturate around 32 rounds.
>
> |Expansion Rounds|OSWorld Succ.|All-Pass@4|Incremental Cost (%)|
> |-|-|-|-|
> |0 (4 PS only)|15.8|7.1|Baseline|
> |8|17.8|10.2|+25%|
> |16|20.0|12.8|+50%|
> |32|21.4|14.1|+100%|
>
> **Efficient approximation**:
>
> Based on the above analysis, RoTS can also be used in reduced-budget settings:
> - Fewer expansion rounds: 16 rounds halve the incremental cost of the standard setting, while achieving +26.6% gain on OSWorld over the PS-only baseline.
> - RoTS can also be adopted in a self-training pipeline with 1 policy model, which can reduce the policy-rollout cost substantially.
> - FDE and EIR need not both be enabled. Under tighter budgets, either module improves over the PS-only baseline (Table 4), while enabling both yields the best performance.
>
> ## Q3: Ablation over error horizon
> This is a good suggestion. Our paper provides partial evidence: Fig. 5(a) shows more expansion rounds improve performance across all error depths and robustness metrics. However, these do not isolate the recovery horizon itself.
>
> To address this, we add a direct depth-cap ablation under the 100k/7B setting: filtering Dref by maximum recovery depth while keeping total budget and λ_ref=0.1 fixed.
>
> |Max Recovery Depth|GUI-RobustEval Post. Succ.|All-Pass@4|OSWorld Succ.|
> |-|-|-|-|
> |No Dref|12.1|8.0|18.5|
> |≤2|15.5|10.5|19.5|
> |≤5|20.7|13.4|21.1|
> |≤7|21.5|13.8|21.3|
> |W/o depth-cap|22.1|14.1|21.4|
>
> The main takeaway is that the largest marginal gain comes from moderate-depth recovery data (especially ≤5), while gains beyond depth 7 are small. This suggests that RoTS does benefit from long-horizon recovery supervision, but the most cost-effective region is moderate depth.
>
> ## Q4: Baseline coverage
> We appreciate this suggestion and agree that the paper should position itself more clearly against recent long-horizon agent work. WebExplorer focuses on web-task synthesis, while UI-NEXUS, CES, and AndroTMem improve execution via scheduling, multi-agent control, or memory. In contrast, RoTS is a training-time, data-centric method for improving recovery from policy-induced errors.
>
> A direct comparison on GUI-RobustEval is challenging because these methods target different settings (web, mobile, vs. multimodal desktop), and our controlled prefix replay is naturally suited to standalone policies but less directly applicable to system-level methods with scheduler or memory states. We therefore view them as largely complementary: RoTS improves the base policy through training data, while those methods strengthen inference-time execution. We will clarify this positioning in the revision.

---

> > ### Author Rebuttal · Reviewer_g97p · 2026-04-03
> >
> > The rebuttal gives additional experiment results on qwen3 comparison and more ablations, while also discussing the computational cost breakdown. My previous questions are fully resolved. I will keep my positive recommendation.

---

### Decision · Program_Chairs · 2026-04-30

**Decision:**

Accept (spotlight)

**Comment:**

This paper addresses an important question of building GUI agents (their inability to recover from errors caused by themselves), which has been overlooked in existing benchmarks. All reviewers vote strongly for the paper. I think this paper would greatly benefit the community. It's a timely contribution on a timely topic.